# Minimax Value Interval for Off-Policy Evaluation and Policy Optimization

**Nan Jiang**
Department of Computer Science
University of Illinois at Urbana-Champaign
Urbana, IL 61801
`nanjiang@illinois.edu`

**Jiawei Huang**
Department of Computer Science
University of Illinois at Urbana-Champaign
Urbana, IL 61801
`jiaweih@illinois.edu`

## Abstract

We study minimax methods for off-policy evaluation (OPE) using value functions and marginalized importance weights. Despite that they hold promises of overcoming the exponential variance in traditional importance sampling, several key problems remain:
(1) They require function approximation and are generally biased. For the sake of trustworthy OPE, is there anyway to quantify the biases?
(2) They are split into two styles ("weight-learning" vs "value-learning"). Can we unify them?
In this paper we answer both questions positively. By slightly altering the derivation of previous methods (one from each style [1]), we unify them into a single *value interval* that comes with a special type of double robustness: when *either* the value-function or the importance-weight class is well specified, the interval is valid and its length quantifies the misspecification of the other class. Our interval also provides a unified view of and new insights to some recent methods, and we further explore the implications of our results on exploration and exploitation in off-policy policy optimization with insufficient data coverage.

## 1 Introduction

A major barrier to applying reinforcement learning (RL) to real-world applications is the difficulty of evaluation: how can we reliably evaluate a new policy before actually deploying it, possibly using historical data collected from a different policy? Known as off-policy evaluation (OPE), the problem is genuinely difficult as the variance of any unbiased estimator—including the popular importance sampling methods and their variants [2, 3]—inevitably grows exponentially in horizon [4].

To overcome this "curse of horizon", the RL community has recently gained interest in a new family of algorithms [e.g., 5], which require function approximation of value-functions and marginalized importance weights and provide accurate evaluation when *both* function classes are well-specified (or *realizable*). Despite that fast progress is made in this direction, several key problems remain:

- The methods are generally biased since they rely on function approximation. Is there anyway we can quantify the biases, which is important for trustworthy evaluation?

- The original method by Liu et al. [5] estimates the marginalized importance weights ("weight") using a discriminator class of value-functions ("value"). Later, Uehara et al. [1] swap the roles of value and weight to learn a $Q$-function using weight discriminators. Not only we have two styles of methods now ("weight-learning" vs "value-learning"), each of them also ignores some important components of the data in their core optimization (see Sec. 3 for details). Can we have a unified method that makes effective use of all components of data?

In this paper we answer both questions positively. By modifying the derivation of one method from each style [1], we unify them into a single *value interval*, which automatically comes with a special type of double robustness: when either the weight- or the value-class is well specified, the interval is valid and its length quantifies the misspecification of the other class (Sec. 4). Each bound is computed from a single optimization program that uses all components of the data, which we show is generally tighter than the naïve intervals developed from previous methods. Our derivation also unifies several recent OPE methods and reveals their simple and direct connections; see Table 1 in the appendix. Furthermore, we examine the potential of applying these value bounds to two long-standing problems in RL: reliable off-policy policy optimization under poor data coverage (i.e., *exploitation*), and efficient *exploration*. Based on a simple but important observation, that poor data coverage can be treated as a special case of importance-weight misspecification, we show that optimizing our lower and upper bounds over a policy class corresponds to the well-established pessimism and optimism principles for these problems, respectively (Sec. 5).

**On Statistical Errors** We assume exact expectations and ignore statistical errors in most of the derivations, as our main goal is to quantify the biases and unify existing methods. In Appendix L we show how to theoretically handle the statistical errors by adding generalization error bounds to the interval. That said, these generalization bounds are typically loose for practical purposes, and we handle statistical errors by bootstrapping in the experiments (Section 4.4) and show its effectiveness empirically. We also refer the readers to concurrent works that provide tighter and/or more efficient computation of confidence intervals for related estimators [6, 7].

## 2 Preliminaries

**Markov Decision Processes** An infinite-horizon discounted MDP is specified by $(\mathcal{S}, \mathcal{A}, P, R, \gamma, s_0)$, where $\mathcal{S}$ is the state space, $\mathcal{A}$ is the action space, $P : \mathcal{S} \times \mathcal{A} \to \Delta(\mathcal{S})$ is the transition function ($\Delta(\cdot)$ is probability simplex), $R : \mathcal{S} \times \mathcal{A} \to \Delta([0, R_{\max}])$ is the reward function, and $s_0$ is a known and deterministic starting state, which is w.l.o.g.[1] For simplicity we assume $\mathcal{S}$ and $\mathcal{A}$ are finite and discrete but their cardinalities can be arbitrarily large. Any policy[2] $\pi : \mathcal{S} \to \mathcal{A}$ induces a distribution of the trajectory, $s_0, a_0, r_0, s_1, a_1, r_1, \ldots$, where $s_0$ is the starting state, and $\forall t \geq 0$, $a_t = \pi(s_t)$, $r_t \sim R(s_t, a_t)$, $s_{t+1} \sim P(s_t, a_t)$. The expected discounted return determines the performance of policy $\pi$, which is defined as $J(\pi) := \mathbb{E}[\sum_{t=0}^{\infty} \gamma^t r_t \mid \pi]$. It will be useful to define the discounted (state-action) occupancy of $\pi$ as $d^\pi(s, a) := \sum_{t=0}^{\infty} \gamma^t d_t^\pi(s, a)$, where $d_t^\pi(\cdot, \cdot)$ is the marginal distribution of $(s_t, a_t)$ under policy $\pi$. $d^\pi$ behaves like an *unnormalized* distribution ($\|d^\pi\|_1 = 1/(1-\gamma)$), and for notational convenience we write $\mathbb{E}_{d^\pi}[\cdot]$ with the understanding that

$$\mathbb{E}_{d^\pi}[f(s, a, r, s')] := \sum_{s \in \mathcal{S}, a \in \mathcal{A}} d^\pi(s, a) \, \mathbb{E}_{r \sim R(s,a), s' \sim P(s,a)}[f(s, a, r, s')].$$

With this notation, the discounted return can be written as $J(\pi) = \mathbb{E}_{d^\pi}[r]$.

The policy-specific $Q$-function $Q^\pi$ satisfies the Bellman equations: $\forall s \in \mathcal{S}, a \in \mathcal{A}$, $Q^\pi(s, a) = \mathbb{E}_{r \sim R(s,a), s' \sim P(s,a)}[r + \gamma Q^\pi(s', \pi)] =: (\mathcal{T}^\pi Q^\pi)(s, a)$, where $Q^\pi(s', \pi)$ is a shorthand for $Q^\pi(s', \pi(s'))$ and $\mathcal{T}^\pi$ is the Bellman update operator. It will be also useful to keep in mind that

$$J(\pi) = Q^\pi(s_0, \pi). \tag{1}$$

**Data and Marginalized Importance Weights** In off-policy RL, we are passively given a dataset and cannot interact with the environment to collect more data. The goal of OPE is to estimate $J(\pi)$ for a given policy $\pi$ using the dataset. We assume that the dataset consists of i.i.d. $(s, a, r, s')$ tuples, where $(s, a) \sim \mu$, $r \sim R(s, a)$, $s' \sim P(s, a)$. $\mu \in \Delta(\mathcal{S} \times \mathcal{A})$ is a distribution from which we draw $(s, a)$, which determines the exploratoriness of the dataset. We write $\mathbb{E}_\mu[\cdot]$ as a shorthand for taking expectation w.r.t. this distribution. The strong i.i.d. assumption is only meant to simplify derivation and presentation, and does not play a crucial role in our results as we do not handle statistical errors.

A concept crucial to our discussions is the *marginalized importance weights*. Given any $\pi$, if $\mu(s, a) > 0$ whenever $d^\pi(s, a) > 0$, define $w_{\pi/\mu}(s, a) := \frac{d^\pi(s,a)}{\mu(s,a)}$. When there exists $(s, a)$ such that $\mu(s, a) = 0$ but $d^\pi(s, a) > 0$, $w_{\pi/\mu}$ does not exist (and hence cannot be realized by any function class). When

it does exist, $\forall f, \mathbb{E}_{d^\pi}[f(s, a, r, s')] = \mathbb{E}_{w_{\pi/\mu}}[f(s, a, r, s')] := \mathbb{E}_\mu[w_{\pi/\mu}(s, a)f(s, a, r, s')]$, where

$$\mathbb{E}_w[\cdot] := \mathbb{E}_\mu[w(s, a) \cdot (\cdot)]$$

is a shorthand we will use throughout the paper, and $\mu$ is omitted in $\mathbb{E}_w[\cdot]$ since importance weights are always applied on the data distribution $\mu$. Finally, OPE would be easy if we knew $w_{\pi/\mu}$, as

$$J(\pi) = \mathbb{E}_{d^\pi}[r] = \mathbb{E}_{w_{\pi/\mu}}[r]. \tag{2}$$

**Function Approximation** Throughout the paper, we assume access to two function classes $\mathcal{Q} \subset (\mathcal{S} \times \mathcal{A} \to \mathbb{R})$ and $\mathcal{W} \subset (\mathcal{S} \times \mathcal{A} \to \mathbb{R})$. To develop intuition, they are supposed to model $Q^\pi$ and $w_{\pi/\mu}$, respectively, though most of our main results are stated without assuming any kind of realizability. We use $\mathcal{C}(\cdot)$ to denote the convex hull of a set. We also make the following compactness assumption so that infima and suprema we introduce later are always attainable.

**Assumption 1.** We assume $\mathcal{Q}$ and $\mathcal{W}$ are compact subsets of $\mathbb{R}^{\mathcal{S} \times \mathcal{A}}$.

## 3  Related Work

**Minimax OPE** Liu et al. [5] proposed the first minimax algorithm for learning marginalized importance weights. When the data distribution (e.g., our $\mu$) covers $d^\pi$ reasonably well, the method provides efficient estimation of $J(\pi)$ without incurring the exponential variance in horizon, a major drawback of importance sampling [2, 4, 3]. Since then, the method has sparked a flurry of interest in the RL community [8, 9, 10, 11, 12, 13, 14, 15].

While most of these methods solve for a weight-function using value-function discriminators and plug into Eq.(2) to form the final estimate of $J(\pi)$, Uehara et al. [1] recently show that one can flip the roles of weight- and value-functions to approximate $Q^\pi$. This is also closely related to the kernel loss for solving Bellman equations [16]. As we will see, when we keep model misspecification in mind and derive upper and lower bounds of $J(\pi)$ (as opposed to point estimates), the two types of methods merge into almost the same value intervals—except that they are the reverse of each other, and at most one of them is valid in general (Sec. 4.3).

**Drawback of Existing Methods** One drawback of both types of methods is that some important components of the data are ignored in the core optimization. For example, "weight-learning" [e.g., 5, 14] completely ignores the rewards $r$ in its loss function, and only uses it in the final plug-in step. Similarly, "value-learning" [MQL of 1] ignores the initial state $s_0$ until the final plug-in step [see also 16]. In contrast, each of our bounds is computed from a single optimization program that uses all components of the data, and we show the advantage of unified optimization in Table 1 and App. E.

**Double Robustness** Our interval is valid when either function class is well-specified, which can be viewed as a type of double robustness. This is related to but different from the usual notion of double robustness in RL [17, 3, 18, 19], and we explain the difference in App. C.

**AlgaeDICE** Closest related to our work is the recently proposed AlgaeDICE for off-policy policy optimization [20]. In fact, one side of our interval recovers a version of its policy evaluation component. Nachum et al. [20] derive the expression using Fenchel duality [see also 21], whereas we provide an alternative derivation using basic telescoping properties such as Bellman equations (Lemmas 1 and 4). Our results also provide further justification for AlgaeDICE as an off-policy policy optimization algorithm (Sec. 5), and point out its weakness for OPE (Sec. 4) and how to address it.

## 4  The Minimax Value Intervals

In this section we derive the minimax value intervals by slightly altering the derivation of two recent methods [1], one of "weight-learning" style (Sec. 4.1) and one of "value-learning" style (Sec. 4.2), and show that under certain conditions, they merge into a single unified value interval whose validity only relies on either $\mathcal{Q}$ or $\mathcal{W}$ being well-specified (Sec. 4.3). While we focus on the discounted & behavior-agnostic setting in the main text, in Appendix M we describe how to adapt to other settings such as when behavior policy is available or in average-reward MDPs.

### 4.1 Value Interval for Well-specified $\mathcal{Q}$ and Misspecified $\mathcal{W}$

We start with a simple lemma that can be used to derive the "weight-learning" methods, such as the original algorithm by Liu et al. [5] and its behavior-agnostic extension by Uehara et al. [1]. Our derivation assumes realizable $\mathcal{Q}$ but arbitrary $\mathcal{W}$.

**Lemma 1** (Evaluation Error Lemma for Importance Weights). *For any* $w : \mathcal{S} \times \mathcal{A} \to \mathbb{R}$,

$$J(\pi) - \mathbb{E}_w[r] = Q^\pi(s_0, \pi) + \mathbb{E}_w[\gamma Q^\pi(s', \pi) - Q^\pi(s, a)]. \tag{3}$$

*Proof.* By moving terms, it suffices to show that $J(\pi) - Q^\pi(s_0, \pi) = \mathbb{E}_w[r + \gamma Q^\pi(s', \pi) - Q^\pi(s, a)]$, which holds because both sides equal 0. $\qquad\square$

**"Weight-learning" in a Nutshell** The "weight-learning" methods aim to learn a $w$ such that $J(\pi) \approx \mathbb{E}_w[r]$. By Lemma 1, we may simply find $w$ that sets the RHS of Eq.(3) to 0. Of course, this expression depends on $Q^\pi$ which is unknown. However, if we are given a function class $\mathcal{Q}$ that captures $Q^\pi$, we can find $w$ (over a class $\mathcal{W}$) that minimizes

$$\sup_{q \in \mathcal{Q}} |L_{\mathrm{w}}(w, q)|, \quad \text{where } L_{\mathrm{w}}(w, q) := q(s_0, \pi) + \mathbb{E}_w[\gamma q(s', \pi) - q(s, a)]. \tag{4}$$

This derivation implicitly assumes that we can find $w$ such that $L_{\mathrm{w}}(w, q) \approx 0 \; \forall q \in \mathcal{Q}$, which is guaranteed when $w_{\pi/\mu} \in \mathcal{W}$. When $\mathcal{W}$ is misspecified, however, the estimate can be highly biased. Although such a bias can be somewhat quantified by the approximation guarantee of these methods (see Remark 3 for details), we show below that there is a more direct, elegant, and tighter approach.

**Derivation of the Interval** Again, suppose we are given $\mathcal{Q}$ such that $Q^\pi \in \mathcal{Q}$.[3] Then from Lemma 1,

$$J(\pi) = Q^\pi(s_0, \pi) + \mathbb{E}_w[r + \gamma Q^\pi(s', \pi) - Q^\pi(s, a)]$$
$$\leq \sup_{q \in \mathcal{Q}} \left\{ q(s_0, \pi) + \mathbb{E}_w[r + \gamma q(s', \pi) - q(s, a)] \right\}. \tag{5}$$

For convenience, from now on we will use the shorthand

$$L(w, q) := q(s_0, \pi) + \mathbb{E}_w[r + \gamma q(s', \pi) - q(s, a)], \tag{6}$$

and the upper bound is then $\sup_{q \in \mathcal{Q}} L(w, q)$. The lower bound is similar:

$$J(\pi) \geq \inf_{q \in \mathcal{Q}} \left\{ q(s_0, \pi) + \mathbb{E}_w[r + \gamma q(s', \pi) - q(s, a)] \right\} = \inf_{q \in \mathcal{Q}} L(w, q). \tag{7}$$

To recap, an arbitrary $w$ will give us a valid interval $[\inf_{q \in \mathcal{Q}} L(w, q), \; \sup_{q \in \mathcal{Q}} L(w, q)]$, and we may search over a class $\mathcal{W}$ to find a tighter interval[4] by taking the *lowest upper bound* and the *highest lower bound*:

$$J(\pi) \leq \inf_{w \in \mathcal{W}} \sup_{q \in \mathcal{Q}} L(w, q) =: \mathrm{UB}_{\mathrm{w}}, \qquad J(\pi) \geq \sup_{w \in \mathcal{W}} \inf_{q \in \mathcal{Q}} L(w, q) =: \mathrm{LB}_{\mathrm{w}}. \tag{8}$$

It is worth keeping in mind that the above derivation assumes realizable $\mathcal{C}(\mathcal{Q})$. Without such an assumption, there is no guarantee that $J(\pi) \leq \mathrm{UB}_{\mathrm{w}}$, $J(\pi) \geq \mathrm{LB}_{\mathrm{w}}$, or even $\mathrm{LB}_{\mathrm{w}} \leq \mathrm{UB}_{\mathrm{w}}$. Below we establish the conditions under which the interval is valid (i.e., $\mathrm{LB}_{\mathrm{w}} \leq J(\pi) \leq \mathrm{UB}_{\mathrm{w}}$) and tight (i.e., $\mathrm{UB}_{\mathrm{w}} - \mathrm{LB}_{\mathrm{w}}$ is small).

**Properties of the Interval** Intuitively, if $\mathcal{Q}$ is richer, it is more likely to be realizable, which improves the interval's validity. If we further make $\mathcal{W}$ richer, the interval becomes tighter, as we are searching over a richer space to suppress the upper bound and raise the lower bound. We formalize these intuitions with the theoretical results below. Notably, our main results do not require any explicit realizability assumptions and hence are automatically agnostic. All proofs of this section can be found in App. A.

**Theorem 2** (Validity). *Define* $L_{\mathrm{w}}(w, q) := q(s_0, \pi) + \mathbb{E}_w[\gamma q(s', \pi) - q(s, a)]$. *We have*

$$\mathrm{UB}_{\mathrm{w}} - J(\pi) \geq \inf_{w \in \mathcal{W}} \sup_{q \in \mathcal{Q}} L_{\mathrm{w}}(w, q - Q^{\pi}), \qquad J(\pi) - \mathrm{LB}_{\mathrm{w}} \geq \inf_{w \in \mathcal{W}} \sup_{q \in \mathcal{Q}} L_{\mathrm{w}}(w, Q^{\pi} - q).$$

*As a corollary, when $Q^{\pi} \in \mathcal{C}(\mathcal{Q})$, the interval is valid, i.e., $\mathrm{UB}_{\mathrm{w}} \geq J(\pi) \geq \mathrm{LB}_{\mathrm{w}}$.*

$L_{\mathrm{w}}(w, q - Q^{\pi})$ and $L_{\mathrm{w}}(w, Q^{\pi} - q)$ can be viewed as a measure of difference between $Q^{\pi}$ and $q$, as they are essentially linear measurements of $q - Q^{\pi}$ (note that $L_{\mathrm{w}}(w, \cdot)$ is linear) with the measurement vector determined by $w$. Therefore, if $\mathcal{C}(\mathcal{Q})$ contains close approximation of $Q^{\pi}$, then $\mathrm{UB}_{\mathrm{w}}$ will not be too much lower than $J(\pi)$ and $\mathrm{LB}_{\mathrm{w}}$ will not be too much higher than $\mathrm{LB}_{\mathrm{w}}$, and the degree of realizability of $\mathcal{C}(\mathcal{Q})$ determines to what extent $[\mathrm{LB}_{\mathrm{w}}, \mathrm{UB}_{\mathrm{w}}]$ is approximately valid.

Even if valid, the interval may be useless if $\mathrm{UB}_{\mathrm{w}} \gg \mathrm{LB}_{\mathrm{w}}$. Below we show that this can be prevented by having a well-specified $\mathcal{W}$.

**Theorem 3** (Tightness). $\mathrm{UB}_{\mathrm{w}} - \mathrm{LB}_{\mathrm{w}} \leq 2 \inf_{w \in \mathcal{W}} \sup_{q \in \mathcal{Q}} |L_{\mathrm{w}}(w, q)|$. *As a corollary, when $w_{\pi/\mu} \in \mathcal{W}$, we have $\mathrm{UB}_{\mathrm{w}} \leq \mathrm{LB}_{\mathrm{w}}$.*

**Remark 1** (Interpretation of Theorem 3). $\mathrm{UB}_{\mathrm{w}} \leq \mathrm{LB}_{\mathrm{w}}$ does not imply $\mathrm{UB}_{\mathrm{w}} = \mathrm{LB}_{\mathrm{w}}$, as $\mathrm{UB}_{\mathrm{w}}$ and $\mathrm{LB}_{\mathrm{w}}$ may lose their upper/lower bound semantics without realizable $\mathcal{C}(\mathcal{Q})$. An interval that is both tight and valid can only be implied from Theorems 2 and 3 together.

**Remark 2** (Point estimate). If a point estimate is desired, we may output $\frac{1}{2}(\mathrm{UB}_{\mathrm{w}} + \mathrm{LB}_{\mathrm{w}})$, and under $Q^{\pi} \in \mathcal{C}(\mathcal{Q})$ we can assert that its error is bounded by $\inf_{w \in \mathcal{W}} \sup_{q \in \mathcal{Q}} |L_{\mathrm{w}}(w, q)|$. This coincides with the guarantee of MWL under the same assumption [1, Theorem 2]. Furthermore, if we simply output $\mathrm{UB}_{\mathrm{w}}$ or $\mathrm{LB}_{\mathrm{w}}$ (the latter being the policy-evaluation component of Fenchel AlgaeDICE [20])[5], the approximation guarantee will be twice as large since they only incur one-sided errors.

**Remark 3** (Naïve interval). As we alluded to earlier, the approximation guarantee of Uehara et al. [1, Theorem 2] can be used to derive an interval that is also valid under realizable $\mathcal{C}(\mathcal{Q})$; see Table 1 for details. In App. E, we show that such an interval is never tighter than ours under realizable $\mathcal{C}(\mathcal{Q})$.

**Remark 4** (Regularization). In App. F we derive the regularized version of the interval via Fenchel transformations, similar to [20, 21].

## 4.2 Value Interval for Well-specified $\mathcal{W}$ and Misspecified $\mathcal{Q}$

Similar to Sec. 4.1, we now derive the interval for the case of realizable $\mathcal{C}(\mathcal{W})$. We also base our entire derivation on the following simple lemma, which can be used to derive the "value-learning" method.

**Lemma 4** (Evaluation Error Lemma for Value Functions). *For any $q : \mathcal{S} \times \mathcal{A} \to \mathbb{R}$,*

$$J(\pi) - q(s_0, \pi) = \mathbb{E}_{d^{\pi}}[r + \gamma q(s', \pi) - q(s, a)]. \tag{9}$$

*Proof.* $J(\pi) = \mathbb{E}_{d^{\pi}}[r]$, and $\mathbb{E}_{d^{\pi}}[q(s, a) - \gamma q(s', \pi)] = q(s_0, \pi)$ due to Bellman equation for $d^{\pi}$. $\square$

**"Value-learning" in a Nutshell** MQL [1] seeks to find $q$ such that $q(s_0, \pi) \approx J(\pi)$. Using Lemma 4, this can be achieved by finding $q$ that sets the RHS of Eq.(9) to 0, and we can use a class $\mathcal{W}$ that realizes $w_{\pi/\mu}$ to overcome the difficulty of unknown $d^{\pi}$, similar to how we handle the unknown $Q^{\pi}$ in Sec. 4.1. While the method gives an accurate estimation when both $\mathcal{Q}$ and $\mathcal{W}$ are well-specified, below we show how to derive an interval to quantify the bias due to misspecified $\mathcal{Q}$.

**Derivation of the Interval** If we are given $\mathcal{W} \subset (\mathcal{S} \times \mathcal{A} \to \mathbb{R})$ such that $w_{\pi/\mu} \in \mathcal{W}$, then[6]

$$J(\pi) \leq q(s_0, \pi) + \sup_{w \in \mathcal{W}} \mathbb{E}_w[r + \gamma q(s', \pi) - q(s, a)] = \sup_{w \in \mathcal{W}} L(w, q). \tag{10}$$

$$J(\pi) \geq q(s_0, \pi) + \inf_{w \in \mathcal{W}} \mathbb{E}_w[r + \gamma q(s', \pi) - q(s, a)] = \inf_{w \in \mathcal{W}} L(w, q). \tag{11}$$

Again, an arbitrary $q$ yields a valid interval, and we may search over a class $\mathcal{Q}$ to tighten it:

$$J(\pi) \leq \inf_{q \in \mathcal{Q}} \sup_{w \in \mathcal{W}} L(w, q) =: \mathrm{UB}_{\mathrm{q}}, \qquad J(\pi) \geq \sup_{q \in \mathcal{Q}} \inf_{w \in \mathcal{W}} L(w, q) =: \mathrm{LB}_{\mathrm{q}}. \tag{12}$$

**Properties of the Interval** We characterize the interval's validity and tightness similar to Sec. 4.1.

**Theorem 5** (Validity). *Define $L_q(w, q) := \mathbb{E}_w[r + \gamma q(s', \pi) - q(s, a)]$.*

$$\text{UB}_q - J(\pi) \geq \inf_{q \in \mathcal{Q}} \sup_{w \in \mathcal{W}} L_q(w - w_{\pi/\mu}, q), \qquad J(\pi) - \text{LB}_q \geq \inf_{q \in \mathcal{Q}} \sup_{w \in \mathcal{W}} L_q(w_{\pi/\mu} - w, q).$$

*As a corollary, when $w_{\pi/\mu} \in \mathcal{C}(\mathcal{W})$, the interval is valid, i.e., $\text{UB}_w \geq J(\pi) \geq \text{LB}_w$.*

Again, the validity of the interval is controlled by the realizability of $\mathcal{C}(\mathcal{W})$, defined as the best approximation of $w_{\pi/\mu} \in \mathcal{W}$ where the difference between $w_{\pi/\mu}$ and any $w$ is measured by $L_q(w - w_{\pi/\mu}, q)$ and $L_q(w_{\pi/\mu} - w, q)$ (which are linear measurements of $w - w_{\pi/\mu}$).

**Theorem 6** (Tightness). $\text{UB}_q - \text{LB}_q \leq 2 \inf_{q \in \mathcal{Q}} \sup_{w \in \mathcal{W}} |L_q(w, q)|$. *As a corollary, when $Q^\pi \in \mathcal{Q}$, we have $\text{UB}_q \leq \text{LB}_q$.*

### 4.3 Unification

So far we have obtained two intervals:

$$\mathcal{C}(\mathcal{Q}) \text{ realizable:} \begin{cases} \text{UB}_w = \inf_w \sup_q L(w, q), \\ \text{LB}_w = \sup_w \inf_q L(w, q), \end{cases} \qquad \mathcal{C}(\mathcal{W}) \text{ realizable:} \begin{cases} \text{UB}_q = \inf_q \sup_w L(w, q), \\ \text{LB}_q = \sup_q \inf_w L(w, q), \end{cases}$$

where $L(w, q) = q(s_0, \pi) + \mathbb{E}_w[r + \gamma q(s', \pi) - q(s, a)]$. Taking a closer look, $\text{UB}_w$ and $\text{LB}_q$ are almost the same and only differ in the order of optimizing $q$ and $w$, and so are $\text{UB}_q$ and $\text{LB}_w$. It turns out that if $\mathcal{W}$ and $\mathcal{Q}$ are convex, these two intervals are precisely the *reverse* of each other.

**Theorem 7.** *If $\mathcal{W}$ and $\mathcal{Q}$ are compact and convex sets, we have $\text{UB}_w = \text{LB}_q$, $\text{UB}_q = \text{LB}_w$.*

This result implies that we do not need to separately consider these two intervals. Neither do we need to know which one of $\mathcal{Q}$ and $\mathcal{W}$ is well specified. We just need to do the obvious thing, which is to compute $\text{UB}_w(= \text{LB}_q)$ and $\text{UB}_q(= \text{LB}_w)$, and let the smaller number be the lower bound and the greater one be the upper bound. This way we get a single interval that is valid when either $\mathcal{Q}$ or $\mathcal{W}$ is realizable (Theorems 2 and 5), and tight when both are. Furthermore, by looking at which value is lower, we can tell which class is misspecified (assuming one of them is well-specified), and this piece of information may provide guidance to the design of function approximation.

**The Non-convex Case** When $\mathcal{Q}$ and $\mathcal{W}$ are non-convex, the two intervals are still related in an interesting albeit more subtle manner: the two "reversed intervals" are generally tighter than the original intervals, but they are only valid under stronger realizability conditions. See proofs and further discussions in App. B.

**Theorem 8.** *When $Q^\pi \in \mathcal{Q}$, $J(\pi) \in [\text{UB}_q, \text{LB}_q] \subseteq [\text{LB}_w, \text{UB}_w]$. When $w_{\pi/\mu} \in \mathcal{W}$, $J(\pi) \in [\text{UB}_w, \text{LB}_w] \subseteq [\text{LB}_q, \text{UB}_q]$.*

### 4.4 Empirical Verification of Theoretical Predictions

We provide preliminary empirical results to support the theoretical predictions, that
(1) which bound is the upper bound depends on the expressivity of function classes (Sec. 4.3), and
(2) our interval is tighter than the naïve intervals based on previous methods (App.E).
We conduct the experiments in CartPole, with the target policy being softmax over a pre-trained $Q$-function with temperature $\tau$ (behavior policy is $\tau = 1.0$). We use neural nets for $\mathcal{Q}$ and $\mathcal{W}$, and optimize the losses using stochastic gradient descent ascent (SGDA);[7] see App. G for more details. Fig. 1 demonstrates the interval reversal phenomenon, where we compute $\text{LB}_q(\approx \text{UB}_w)$ and $\text{UB}_q(\approx \text{LB}_w)$ for Q-networks of different sizes while fixing everything else. As predicted by theory, $\text{UB}_q > \text{LB}_q$ when $\mathcal{Q}$ is small (hence poorly specified), and the interval is flipped as $\mathcal{Q}$ increases.

Fig. 3 (left) compares the interval lengths between our interval and that induced by MQL under different sample sizes,[8] and we see the former is significantly tighter than the latter. However, Fig. 3

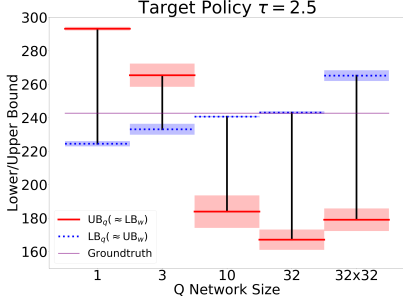

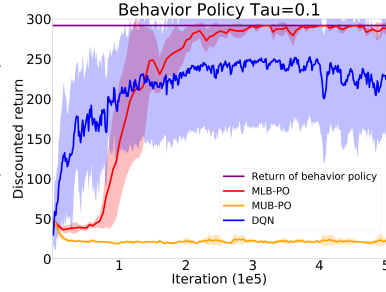

Figure 1: **(Left)** The interval reversal phenomenon; see Sec. 4.4.

Figure 2: **(Right)** Policy optimization under non-exploratory data; see Sec. 5.3 for details.

Figure 3: Comparison of interval length and validity ratio between our interval and that of MQL as a function of sample size; see App. G.3 for more details. All error bars show twice standard errors.

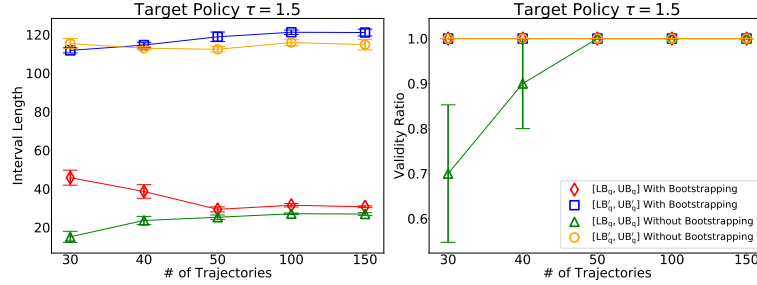

(right) reveals that when the sample size is small, our new interval is too aggressive and fails to contain $J(\pi)$ with a significant chance. This is also expected as our theory does not handle statistical errors, and validity guarantee can be violated due to data randomness especially in the small sample regime. While a theoretical investigation of the statistical errors (beyond the use of generalization error bounds in App. L which can be loose) is left to future work, we empirically test a popular heuristic of bootstrapping intervals (App.G.3), and show that after bootstrapping our interval becomes 100% valid, while its length has only increased moderately especially in the large sample regime.

# 5 On Policy Optimization with Insufficient Data Coverage

OPE is not only useful as an evaluation method, but can also serve as a crucial component for off-policy policy optimization: if we can estimate the return $J(\pi)$ for a class of policies $\Pi$, then in principle we may use $\arg\max_{\pi \in \Pi} J(\pi)$ to find the best policy in class. However, our interval gives two estimations of $J(\pi)$, an upper bound and a lower bound. Which one should we optimize? Traditional wisdom in robust MDP literature [e.g., 23] suggests *pessimism*, i.e., optimizing the lower bound for the best worst-case guarantees. *But which side of the interval is the lower bound?*

As Sec.4.3 indicates, every expression could be the upper bound or the lower bound, depending on the realizability of $\mathcal{Q}$ and $\mathcal{W}$. So the question becomes: which one of the function classes is more likely to be misspecified?

**RL with Insufficient Data Coverage** Even if $\mathcal{W}$ and $\mathcal{Q}$ are chosen with maximum care, the realizability of $\mathcal{W}$ faces one additional and crucial challenge compared to $\mathcal{Q}$: if the data distribution $\mu$ itself does not cover the support of $d^\pi$, then $w_{\pi/\mu}$ may not even exist and hence $\mathcal{W}$ will be poorly specified whatsoever. Since the historical dataset in real applications often comes with no exploratory guarantees, this scenario is highly likely and remains as a crucial yet understudied challenge [24, 11]. In this section we analyze the algorithms that optimize the upper and the lower bounds when $\mathcal{W}$ is poorly specified. For convenience we will call them[9]

$$\text{MUB-PO: } \arg\max_{\pi \in \Pi} \text{UB}_w^\pi, \qquad \text{MLB-PO: } \arg\max_{\pi \in \Pi} \text{LB}_w^\pi, \qquad (13)$$

where MUB-PO/MLB-PO stand for minimax upper (lower) bound policy optimization. MLB-PO is essentially Frenchel AlgaeDICE [20], so our results provide further justification for this method. On the other hand, while MUB-PO may not be appropriate for exploitation, it exercises *optimism* and may be better applied to induce exploration, that is, collecting new data with the policy computed by MUB-PO may improve the data coverage.

## 5.1 Case Study: Exploration and Exploitation in a Tabular Scenario

To substantiate the claim that MUB-PO/MLB-PO induce effective exploration/exploitation, we first consider a scenario where the solution concepts for exploration and exploitation are clearly understood, and show that MUB-PO/MLB-PO reduce to familiar algorithms.

**Setup** Consider an MDP with finite and discrete state and action spaces. Let $\tilde{\mathcal{S}} \subset \mathcal{S}$ be a subset of the state space, and $s_0 \in \tilde{\mathcal{S}}$. Suppose the dataset contains $n$ transition samples from each $s \in \tilde{\mathcal{S}}, a \in \mathcal{A}$ with $n \to \infty$ (i.e., the transitions and rewards in $\tilde{\mathcal{S}}$ are fully known), and 0 samples from $s \notin \tilde{\mathcal{S}}$.

**Known Solution Concepts** In such a simplified setting, effective exploration can be achieved by (the episodic version of) the well-known Rmax algorithm [25, 26], which computes the optimal policy of the Rmax-MDP: this MDP has the same transition dynamics and reward function as the true MDP $M$ on states with sufficient data ($\tilde{\mathcal{S}}$), and the "unknown" states are assumed to have self-loops with $R_{\max}$ rewards, making them appealing to visit and thus encouraging exploration. Similarly, when the goal is to output a policy with the best worst-case guarantee (exploitation), one simply changes $R_{\max}$ to the minimally possible reward ("$R_{\min}$", which is 0 for us). Below we show that MUB-PO and MLB-PO in this setting precisely correspond to the Rmax and the Rmin algorithms, respectively.

**Proposition 9.** *Consider the MDP and the dataset described above. Let $\mathcal{W} = [0, \frac{|\tilde{\mathcal{S}} \times \mathcal{A}|}{1-\gamma}]^{\mathcal{S} \times \mathcal{A}}$ and $\mathcal{Q} = [0, R_{\max}/(1-\gamma)]^{\mathcal{S} \times \mathcal{A}}$. In this case, MUB-PO reduces to Rmax, and MLB-PO reduces to Rmin.*

Despite the intuitiveness of the statement, the proof is quite involved and requires repeated applications of the results established in Sec. 4; we refer interested readers to App. I for proof details.

## 5.2 Guarantees in the Function Approximation Setting

We give more general guarantee in the function approximation setting. For simplicity we do not consider e.g., approximation/estimation errors, and incorporating them is routine [27, 28, 29].

**Exploitation with Well-specified $\mathcal{Q}$** We start with the guarantee of MLB-PO.

**Proposition 10** (MLB-PO). *Let $\Pi$ be a policy class, and assume $Q^\pi \in \mathcal{C}(\mathcal{Q}) \; \forall \pi \in \Pi$. Let $\hat{\pi} = \arg\max_{\pi \in \Pi} \mathrm{LB}_w^\pi$. Then, for any $\pi \in \Pi$, $J(\hat{\pi}) \geq \mathrm{LB}_w^\pi$. As a corollary, for any $\pi$ s.t. $w_{\pi/\mu} \in \mathcal{W}$, $J(\hat{\pi}) \geq J(\pi)$, that is, we compete with any policy whose importance weight is realized by $\mathcal{W}$.*

**Exploration with (Less) Well-specified $\mathcal{Q}$** We then provide the exploration guarantee of MUB-PO, which is an "optimal-or-explore" statement, that either the obtained policy is near-optimal (to be defined below), or it will induce effective exploration. Perhaps surprisingly, our results suggest that MUB-PO for exploration might be significantly more robust against misspecified $\mathcal{Q}$ than MLB-PO for exploitation.

The key idea behind the agnostic result is the following: instead of competing with $\max_{\pi \in \Pi} J(\pi)$ as the optimal value under the assumption that $Q^\pi \in \mathcal{C}(\mathcal{Q}), \forall \pi \in \Pi$ (the same assumption as MLB-PO), we aim at a less ambitious notion of optimality under a substantially relaxed assumption; without any explicit assumption on $\mathcal{Q}$, we directly compete with $\max_{\pi \in \Pi: Q^\pi \in \mathcal{C}(\mathcal{Q})} J(\pi)$. In words, we compete with any policy whose Q-function is realized by $\mathcal{C}(\mathcal{Q})$. When $Q^\pi \in \mathcal{C}(\mathcal{Q})$ for $\pi = \arg\max_{\pi \in \Pi} J(\pi)$, we compete with the usual notion of optimal value. However, even if some (or most) policies' Q-functions elude $\mathcal{C}(\mathcal{Q})$, we can still compete with whichever policy whose Q-function is captured by $\mathcal{C}(\mathcal{Q})$. A similar notion of optimality has been used by Jiang et al. [30], and indeed their algorithm is closely related to MUB-PO, which we discuss in App. J.4.

We state a short version of MUB-PO's guarantee, with the full version deferred to App. J.

**Proposition 11** (MUB-PO, short ver.). *Let $\Pi$ be a policy class. Let $\hat{\pi} = \arg\max_{\pi \in \Pi} \mathrm{UB}_w^\pi$. Assuming $\|q\|_\infty \leq R_{\max}/(1-\gamma) \; \forall q \in \mathcal{Q}$, we have for any $w \in \mathcal{W}$,*

$$\|w \cdot \mu - d^{\hat{\pi}}\|_1 \geq \frac{(1-\gamma)\left(\max_{\pi \in \Pi: Q^\pi \in \mathcal{C}(\mathcal{Q})} J(\pi) - J(\hat{\pi})\right)}{2R_{\max}}.$$

Recall that $w \in \mathcal{W}$ is supposed to model the importance weight that coverts data $\mu$ to the occupancy of some policy, e.g., $w_{\pi/\mu} \cdot \mu = d^\pi$. The proposition states that either $\hat{\pi}$ is near-optimal, or it will induce an occupancy that cannot be accurately modeled by *any* importance weights in $\mathcal{W}$ when

applied on the current data distribution $\mu$. Hence, if $\mathcal{W}$ is very rich and models all distributions covered by $\mu$, then $\hat{\pi}$ must visit new state-actions or it must be near-optimal.

### 5.3  Preliminary Empirical Results

While we would like to test MLB-PO and MUB-PO in experiments, the joint optimization of $\pi, w, q$ is very challenging and remains an open problem [20]. Similar to Nachum et al. [20], we try a heuristic variant of MLB-PO and MUB-PO that works in near-deterministic environments; see App. K for details. As Fig. 5 shows, when the behavior policy is non-exploratory ($\tau = 0.1$ yields a nearly deterministic policy), MLB-PO can reliably achieve good performance despite the lack of explicit regularization towards the behavior policy, whereas DQN is more unstable and suffers higher variance. MUB-PO, on the other hand, fails to achieve a high value—which is also expected from theory—but is able to induce exploration in certain cases. We defer more detailed results and discussions to App. K due to space limit.

## 6  Conclusions and Open Problems

We derive a minimax value interval for off-policy evaluation. The interval is valid as long as either the importance-weight or the value-function class is well specified, and its length quantifies the misspecification error of the other class. Our highly simplified derivations only take a few steps from the basic Bellman equations, which condense and unify the derivations in existing works. When applied to off-policy policy optimization in face of insufficient data coverage, which is an important scenario of practical concerns, optimizing our lower and upper bounds over a policy class can induce effective exploitation and exploration, respectively.

We conclude the paper with open problems for future work:

- We handled sampling errors via bootstrapping in the experiments. Are there statistically more effective and computationally more efficient solutions?

- MLB-PO and MUB-PO exhibit promising statistical properties but require difficult optimization. Can we develop principled and practically effective strategies for optimizing these objectives?

- The double robustness of our interval protects against the misspecification of either $\mathcal{Q}$ or $\mathcal{W}$, but still requires one of them to be realizable to guarantee validity. Is it possible at all to develop an interval that is *always* valid, and whose length is allowed to depend on the misspecification errors of both $\mathcal{Q}$ and $\mathcal{W}$? If impossible, can we establish information-theoretic hardness, and does reformulating the problem in a practically relevant manner help circumvent the difficulty?

Answering these questions will be important steps towards reliable and practically useful off-policy evaluation.

## Broader Impact

This work is largely of theoretical nature, trying to unify existing methods and pointing out their connections, with minimal proof-of-concept simulation experiments. Therefore, we do not foresee direct broader impact. That said, an important motivation for this work is to equip RL with off-line evaluation methods that rely on as few assumptions as possible, and in the long term this should contribute to a more trustworthy framework for applying RL to real-world tasks, where reliable evaluation is indispensable. We warn, however, that even though we aim at a less ambitious goal of producing a valid interval (whose length may not go to $0$ as sample size increases), we still require unverifiable assumptions (realizability of $\mathcal{C}(\mathcal{Q})$ or $\mathcal{C}(\mathcal{W})$). Therefore, the value intervals produced by this and subsequent papers should be interpreted and treated with care and not taken as-is in application scenarios. There are also several important aspects of building practically useful confidence intervals that are ignored in this paper (since we are still in the early stage of theoretical investigations), such as the handling of statistical errors and possible confoundedness in the data, which need to be addressed by future works before these methods can be readily deployed in applications.

## Acknowledgments and Disclosure of Funding

This project is partially supported by a Microsoft Azure University Grant. The authors thank Jinglin Chen for pointing out several mistakes/typos in an earlier draft of the paper.

## Footnotes

[1] When $s_0 \sim d_0$ is random, all our derivations hold by replacing $q(s_0, \pi)$ with $\mathbb{E}_{s_0 \sim d_0}[q(s_0, \pi)]$.

[2] Our derivation also applies to stochastic policies.

[3]This condition can be relaxed to $Q^\pi \in \mathcal{C}(\mathcal{Q})$ due to the affinity of $L(w, \cdot)$.

[4]We cannot search over the unrestricted (tabular) class when the state space is large, due to overfitting. In contrast, the value bounds derived in this paper only optimize over restricted $\mathcal{Q}$ and $\mathcal{W}$ classes, allowing the bound computed on a finite sample to generalize when $\mathcal{Q}$ and $\mathcal{W}$ have bounded statistical complexities; see Appendix L for related discussions.

[5]See Appendix D for how to translate between the two papers' notations.

[6]As before, this condition can be relaxed to $w_{\pi/\mu} \in \mathcal{C}(\mathcal{W})$.

[7]Neural nets induce non-convex classes, violating Theorem 7's assumptions. However, such non-convexity may be mitigated by having an ensemble of neural nets, or in the recently popular infinite-width regime [22]. Also, SGDA is symmetric w.r.t. $w$ and $q$, and our implementation can be viewed as heuristic approximations of $\text{UB}_q$ & $\text{LB}_w$ and $\text{UB}_w$ & $\text{LB}_q$, resp., so we treat $\text{UB}_q \approx \text{LB}_w$ and $\text{UB}_w \approx \text{LB}_q$ in Sec. 4.4.

[8]We compare to the interval induced by MQL ($[\text{LB}_q', \text{UB}_q']$ in Table 1), as Uehara et al. [1] reported that MQL is more stable and performs better than MWL in discounted problems.

[9]We make the dependence of $\text{UB}_w$ and $\text{LB}_w$ on $\pi$ explicit since we consider multiple policies in this section.

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
