[Supplementary Material]

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

 $\mathrm{UB_q}$ & $\mathrm{LB_w}$ and $\mathrm{UB_w}$ & $\mathrm{LB_q}$, resp., so we treat $\mathrm{UB_q} \approx \mathrm{LB_w}$ and $\mathrm{UB_w} \approx \mathrm{LB_q}$ in Sec. 4.4.

[8] We compare to the interval induced by MQL ($[\mathrm{LB_q}', \mathrm{UB_q}']$ in Table 1), as Uehara et al. [1] reported that MQL is more stable and performs better than MWL in discounted problems.

[9]We make the dependence of $\text{UB}_\text{w}$ and $\text{LB}_\text{w}$ on $\pi$ explicit since we consider multiple policies in this section.

[10]Point-wise supremum is lower semi-continuous, i.e., $\sup_{q \in \mathcal{Q}} L(w,q)$ is lower semi-continuous in $w$ and hence the infimum is attainable.

[11] This is because $L_{\mathrm{q}}(w, q)$ is linear in $w$, and Eq.(17) becomes an identity.

[12]Many policies are indistinguishable under the original 0/1 reward function, so we define an angle-dependent reward function that takes numerical values. We also add random noise to make the transitions stochastic.

[13]https://github.com/openai/baselines

[14]With a slight abuse of notations, in this section we treat $R : \mathcal{S} \times \mathcal{A} \to [0, R_{\max}]$ as a deterministic reward function for convenience.

[15] $w \cdot \mu$ may be unnormalized, but this does not affect our results.

[16]The original OLIVER algorithm requires the approximation error of $\mathcal{Q}$ as an input, which can be avoided by replacing its constrained optimization step with an unconstrained one similar to MUB-PO.

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

Table 1: Comparison of the upper and lower bounds of $J(\pi)$ obtained by different OPE methods. Rows 1,2,4,5 are the bounds derived in this paper, and Rows 3,6 (with $'$) are their naïve counterparts obtained by the approximation guarantees of previous methods, which are looser (see Appendix E). $\hat{w}$ (and $\hat{q}$) is the $w$ (and $q$) that attains the infimum in the rest of the expression, e.g., $\hat{w} := \arg\min_w \sup_q |L_w(w,q)|$. The **red** terms highlight why the naïve bounds are loose. All expressions with subscript "w" are valid upper or lower bounds when $\mathcal{C}(\mathcal{Q})$ is realizable, and those with "q" are derived assuming realizable $\mathcal{C}(\mathcal{W})$.

$L_w(w,q) := q(s_0,\pi) + \mathbb{E}_w[\gamma q(s',\pi) - q(s,a)]$.    $L_q(w,q) := \mathbb{E}_w[r + \gamma q(s',\pi) - q(s,a)]$.
Our loss $L(w,q) = \mathbb{E}_w[r] + L_w(w,q) = q(s_0,\pi) + L_q(w,q)$.

|  | Expression | Remark |
|---|---|---|
| $\text{UB}_w$ | $\inf_w \sup_q \left( \mathbb{E}_{\boldsymbol{w}}[\boldsymbol{r}] + L_w(w,q) \right)$ | New |
| $\text{LB}_w$ | $\sup_w \inf_q \left( \mathbb{E}_{\boldsymbol{w}}[\boldsymbol{r}] + L_w(w,q) \right)$ | Fenchel AlgaeDICE [20] |
| $\text{UB}_w{}'$ $\text{LB}_w{}'$ | $\mathbb{E}_{\hat{\boldsymbol{w}}}[\boldsymbol{r}] \pm \inf_w \sup_q |L_w(w,q)|$ | MWL [1] see also Liu et al. [5], Zhang et al. [14] |
| $\text{UB}_q$ | $\inf_q \sup_w \left( \boldsymbol{q(s_0,\pi)} + L_q(w,q) \right)$ | $= \text{LB}_w$ with convex $\mathcal{Q}$ and $\mathcal{W}$ |
| $\text{LB}_q$ | $\sup_q \inf_w \left( \boldsymbol{q(s_0,\pi)} + L_q(w,q) \right)$ | $= \text{UB}_w$ with convex $\mathcal{Q}$ and $\mathcal{W}$ |
| $\text{UB}_q{}'$ $\text{LB}_q{}'$ | $\boldsymbol{\hat{q}(s_0,\pi)} \pm \inf_q \sup_w |L_q(w,q)|$ | MQL [1] |

# A  Proofs of Section 4

**Proof of Theorem 2.** Let $w_1$ denote the $w$ that attains the infimum in $\text{UB}_w$.[10]

$$\text{UB}_w - J(\pi) = \sup_{q \in \mathcal{Q}} L(w_1,q) - L(w_1, Q^\pi) \qquad \text{(Lemma 1: } \forall w, J(\pi) = L(w, Q^\pi)\text{)}$$

$$\geq \inf_{w \in \mathcal{W}} \sup_{q \in \mathcal{Q}} \{L(w,q) - L(w, Q^\pi)\}$$

$$= \inf_{w \in \mathcal{W}} \sup_{q \in \mathcal{Q}} L_w(w, q - Q^\pi). \qquad (\mathbb{E}_w[r] \text{ terms cancel})$$

Similarly, let $w_2$ denote the $w$ that attains the supremum in $\text{LB}_w$,

$$J(\pi) - \text{LB}_w = L(w_2, Q^\pi) - \inf_{q \in \mathcal{Q}} L(w_2, q) \geq \inf_{w \in \mathcal{W}} \sup_{q \in \mathcal{Q}} \{L(w, Q^\pi) - L(w,q)\}.$$

The corollary follows because $\forall w$, both $\sup_{q \in \mathcal{Q}}\{L(w,q) - L(w, Q^\pi)\}$ and $\sup_{q \in \mathcal{Q}}\{L(w, Q^\pi) - L(w,q)\}$ are non-negative when $Q^\pi \in \mathcal{C}(\mathcal{Q})$, noting that $L(w, \cdot)$ is affine. $\square$

**Proof of Theorem 3.**

$$\text{UB}_w - \text{LB}_w = \inf_{w \in \mathcal{W}} \sup_{q \in \mathcal{Q}} L(w,q) - \sup_{w \in \mathcal{W}} \inf_{q \in \mathcal{Q}} L(w,q)$$

$$= \inf_{w,w' \in \mathcal{W}} \left\{ \sup_{q \in \mathcal{Q}} L(w,q) - \inf_{q \in \mathcal{Q}} L(w',q) \right\}$$

$$\leq \inf_{w \in \mathcal{W}} \left\{ \sup_{q \in \mathcal{Q}} L(w,q) - \inf_{q \in \mathcal{Q}} L(w,q) \right\}. \qquad \text{(constraining } w = w'\text{)}$$

Noting that $L(w,q) = \mathbb{E}_w[r] + L_w(w,q)$,

$$\sup_{q \in \mathcal{Q}} L(w,q) - \inf_{q \in \mathcal{Q}} L(w,q) = \sup_{q \in \mathcal{Q}} L_w(w,q) - \inf_{q \in \mathcal{Q}} L_w(w,q) \leq 2 \sup_{q \in \mathcal{Q}} |L_w(w,q)|.$$

When $w_{\pi/\mu} \in \mathcal{W}$, $\text{UB}_w \leq \text{LB}_w$ follows from the fact that $L_w(w_{\pi/\mu}, q) \equiv 0, \forall q$. $\square$

***Proof of Theorem 5***. Let $q_1$ denote the $q$ that attains the infimum in $\mathrm{UB_q}$.

$$\mathrm{UB_q} - J(\pi) = \sup_{w \in \mathcal{W}} L(w, q_1) - L(w_{\pi/\mu}, q_1) \qquad\qquad (\forall q,\, J(\pi) = L(w_{\pi/\mu}, q))$$

$$\geq \inf_{q \in \mathcal{Q}} \sup_{w \in \mathcal{W}} \{L(w, q) - L(w_{\pi/\mu}, q)\}.$$

Similarly, let $q_2$ denote the $q$ that attains the supremum in $\mathrm{LB_q}$,

$$J(\pi) - \mathrm{LB_q} = L(w_{\pi/\mu}, q_2) - \inf_{w \in \mathcal{W}} L(w, q_2) \geq \inf_{q \in \mathcal{Q}} \sup_{w \in \mathcal{W}} \{L(w_{\pi/\mu}, q) - L(w, q)\}.$$

The corollary follows because $\forall q$, both $\sup_{w \in \mathcal{C}(\mathcal{W})}\{L(w, q) - L(w_{\pi/\mu}, q)\}$ and $\sup_{w \in \mathcal{C}(\mathcal{W})}\{L(w_{\pi/\mu}, q) - L(w, q)\}$ are non-negative when $w_{\pi/\mu} \in \mathcal{C}(\mathcal{W})$. ☐

***Proof of Theorem 6***.

$$\mathrm{UB_q} - \mathrm{LB_q} = \inf_{q \in \mathcal{Q}} \sup_{w \in \mathcal{W}} L(w, q) - \sup_{q \in \mathcal{Q}} \inf_{w \in \mathcal{W}} L(w, q)$$

$$= \inf_{q, q' \in \mathcal{Q}} \left\{ \sup_{w \in \mathcal{W}} L(w, q) - \inf_{w \in \mathcal{W}} L(w, q') \right\}$$

$$\leq \inf_{q \in \mathcal{Q}} \left\{ \sup_{w \in \mathcal{W}} L(w, q) - \inf_{w \in \mathcal{W}} L(w, q) \right\}. \qquad\qquad (\text{constraining } q = q')$$

Note that $L(w, q) = q(s_0, \pi) + L_\mathrm{q}(w, q)$, therefore,

$$\sup_{w \in \mathcal{W}} L(w, q) - \inf_{w \in \mathcal{W}} L(w, q) = \sup_{w \in \mathcal{W}} L_\mathrm{q}(w, q) - \inf_{w \in \mathcal{W}} L_\mathrm{q}(w, q) \leq 2 \sup_{w \in \mathcal{W}} |L_\mathrm{q}(w, q)|.$$

When $Q^\pi \in \mathcal{Q}$, $\mathrm{UB_q} \leq \mathrm{LB_q}$ follows from the fact that $L_\mathrm{q}(w, Q^\pi) \equiv 0, \forall w$. ☐

***Proof of Theorem 7***. Since $L(w, q)$ is bi-affine—which implies that it is convex-concave—by the minimax theorem [31, 32], the order of $\inf$ and $\sup$ are exchangeable. ☐

# B  Relationships between the Intervals in the Non-convex Case

When $\mathcal{Q}$ and $\mathcal{W}$ are non-convex, Theorem 7 no longer holds but the intervals derived in Sec. 4.1 and 4.2 are still related in an interesting albeit more subtle manner; as stated in Theorem 8, the two "reversed intervals" are generally tighter than the original intervals, but they are only valid under stronger realizability conditions.

Below we provide the proof of Theorem 8.

***Proof of Theorem 8***. We only prove the first statement under $Q^\pi \in \mathcal{Q}$, and the proof for the second statement under $w_{\pi/\mu} \in \mathcal{W}$ is similar and omitted. First of all, $[\mathrm{UB_q}, \mathrm{LB_q}] \subseteq [\mathrm{LB_w}, \mathrm{UB_w}]$ holds without assuming $Q^\pi \in \mathcal{Q}$, as $\sup \inf(\cdot) \leq \inf \sup(\cdot)$. Hence it suffices to show $J(\pi) \in [\mathrm{UB_q}, \mathrm{LB_q}]$.

To prove this statement, we go back to Lemma 1, which states $J(\pi) = L(w, Q^\pi)$ holds for arbitrary $w$. Therefore,

$$\sup_{w \in \mathcal{W}} L(w, Q^\pi) = J(\pi) = \inf_{w \in \mathcal{W}} L(w, Q^\pi).$$

Now if we have $\mathcal{Q}$ such that $Q^\pi \in \mathcal{Q}$, we immediately have

$$\inf_{q \in \mathcal{Q}} \sup_{w \in \mathcal{W}} L(w, q) \leq \sup_{w \in \mathcal{W}} L(w, Q^\pi) = J(\pi) = \inf_{w \in \mathcal{W}} L(w, q) \leq \sup_{q \in \mathcal{Q}} \inf_{w \in \mathcal{W}} L(w, q).$$

This completes the proof. Compared to the derivation in Sec. 4.1, we have changed the order in which $\mathcal{Q}$ and $\mathcal{W}$ are introduced. As a consequence, we need to optimize $q \in \mathcal{Q}$ over the objectives $\sup_{w \in \mathcal{W}} L(w, q)$ and $\inf_{w \in \mathcal{W}} L(w, q)$ which are no longer affine in $q$ (due to $\sup_{w \in \mathcal{W}}$ and $\inf_{w \in \mathcal{W}}$). For this reason, $Q^\pi \in \mathcal{C}(\mathcal{Q})$ is no longer sufficient to guarantee the validity of the interval, and we need the stronger condition $Q^\pi \in \mathcal{Q}$. ☐

In Theorem 8 we provide tighter intervals with stronger realizability assumptions (e.g., $Q^\pi \in \mathcal{Q}$ instead of $Q^\pi \in \mathcal{C}(\mathcal{Q})$). Below we show that such assumptions are necessary, that when we only have $Q^\pi \in \mathcal{C}(\mathcal{Q})$ but not $Q^\pi \in \mathcal{Q}$, the tighter interval $[\mathrm{UB_q}, \mathrm{LB_q}]$ can be invalid. Similar conclusions hold for $[\mathrm{UB_w}, \mathrm{LB_w}]$ which we do not go over in detail.

**Proposition 12.** *There exists an MDP, a target policy $\pi$, a data distribution $\mu$, and function classes $\mathcal{Q}$ and $\mathcal{W}$ satisfying $Q^\pi \notin \mathcal{Q}$ but $Q^\pi \in C(\mathcal{Q})$, where $[\mathrm{UB_q}, \mathrm{LB_q}]$ is invalid.*

*Proof.* We consider the deterministic MDP in Figure 4 with 3 states and 2 actions. $s_0$ is the only initial state, and $s_2$ is an absorbing state. $r(s, a)$ equals 1 if and only if $s = s_0$ and $a = a_0$, otherwise 0. We consider uniform policy $\pi$, i.e. $\pi(s_i|a_j) = 0.5$ for all $i \in \{0, 1, 2\}, j \in \{0, 1\}$. Let $\mu$ be the uniform distribution over $\mathcal{S} \times \mathcal{A}$.

Figure 4: The MDP used in the proof of Proposition 12.

We construct $\mathcal{Q}$ so that it contains two functions $q_1$ and $q_2$, defined as:

$$q_1(s, a) = Q^\pi(s, a) + \epsilon \cdot \mathbb{I}[s = s_1], \qquad q_2(s, a) = Q^\pi(s, a) - \epsilon \cdot \mathbb{I}[s = s_1],$$

for some $\epsilon > 0$. It is easy to verify that $Q^\pi \notin \mathcal{Q}$ but $Q^\pi = \frac{1}{2}(q_1 + q_2) \in C(\mathcal{Q})$. Also let $\mathcal{W}$ be

$$\mathcal{W} = \left\{ \frac{1}{1 - \gamma} \cdot \frac{f(s, a)}{\mathbb{E}_\mu[f(s, a)]} : f \in [0, |\mathcal{S} \times \mathcal{A}|]^{\mathcal{S} \times \mathcal{A}} \right\}$$

We pick out two elements $w_1, w_2$ from $\mathcal{W}$:

$$w_1(s, a) = \frac{6\mathbb{I}[s = s_0, a = a_1]}{1 - \gamma}, \qquad w_2(s, a) = \frac{6\mathbb{I}[s = s_1, a = a_1]}{1 - \gamma}.$$

Let $L(w, q) := q(s_0, \pi) + \mathbb{E}_w[r + \gamma q(s', \pi) - q(s, a)]$, and

$$\sup_{w \in \mathcal{W}} L(w, q_1) - J(\pi) = \sup_{w \in \mathcal{W}} L(w, q_1) - Q^\pi(s_0, \pi) \geq L(w_1, q_1) - Q^\pi(s_0, \pi)$$
$$= q_1(s_0, \pi) + \mathbb{E}_{w_1}[r + \gamma q_1(s', \pi) - q_1(s, a)] - Q^\pi(s_0, \pi)$$
$$= \frac{r(s_0, a_1) + \gamma(\frac{1}{2}q_1(s_1, a_0) + \frac{1}{2}q_1(s_1, a_1)) - q_1(s_0, a_1)}{1 - \gamma}$$
$$= \frac{\gamma \epsilon}{1 - \gamma} > 0.$$

$$\sup_{w \in \mathcal{W}} L(w, q_2) - J(\pi) = \sup_{w \in \mathcal{W}} L(w, q_2) - Q^\pi(s_0, \pi) \geq L(w_2, q_2) - Q^\pi(s_0, \pi)$$
$$= q_2(s_0, \pi) + \mathbb{E}_{w_2}[r + \gamma(q_2(s', \pi)) - q_2(s, a)] - Q^\pi(s_0, \pi)$$
$$= \frac{r(s_1, a_1) + \gamma(\frac{1}{2}q_2(s_2, a_0) + \frac{1}{2}q_2(s_2, a_1)) - q_2(s_1, a_1)}{1 - \gamma}$$
$$= \frac{\epsilon}{1 - \gamma} > 0.$$

Therefore, $\inf_{q \in \mathcal{Q}} \sup_{w \in \mathcal{W}} L(w, q) - J(\pi) = \min\{\sup_{w \in \mathcal{W}} L(w, q_1), \sup_{w \in \mathcal{W}} L(w, q_2)\} > 0$, which implies that the lower bound is invalid. The upper bound can be shown to be invalid in a similar manner. □

## C  On Double Robustness

Our interval is valid when either function class is well-specified, which can be viewed as a type of double robustness. This is related to but different from the usual notion of double robustness in RL [17, 3, 18, 19]: classical doubly robust methods are typically "meta"-estimators and require a value-function whose estimation procedure is unspecified, and the double robustness refers to the fact that the estimation is unbiased and/or enjoys reduced variance if the given value-function is accurate. In comparison, our double robustness gives weaker guarantees (valid interval, as opposed to accurate point estimates) but also requires much weaker assumptions (well-specified function *class* as opposed to an accurate function), so it is important not to confuse the two types of double robustness.

## D  On AlgaeDICE's Notations

In this section, we clarify the difference between the notations in AlgaeDICE [20] and ours.

We take the objective in their Eq.(15) as an example ($\max_\pi$ is dropped):

$$\min_{\nu:\mathcal{S}\times\mathcal{A}\to\mathbb{R}} \max_{\zeta:\mathcal{S}\times\mathcal{A}\to\mathbb{R}} (1-\gamma)\mathbb{E}_{s_0\sim\mu_0,a_0\sim\pi(s_0)}[\nu(s_0,a_0)]$$

$$+ \mathbb{E}_{(s,a)\sim d^D,s'\sim T(s,a),a'\sim\pi(s')}[(\gamma\nu(s',a') + r(s,a) - \nu(s,a))\cdot\zeta(s,a) - \alpha\cdot f(\zeta)].$$

As we can see, if we choose $\alpha = 0$, the above is almost precisely our $\mathrm{UB}_q$ (= $\mathrm{LB}_w$ with convex classes): their $T$ is our $P$; their $\nu$ is our $q$; their $\zeta$ is our $w$; their $\nu(s_0,a_0)$ term corresponds to our $q(s_0,\pi)$ as we assume deterministic initial state w.l.o.g. (see our Footnote 1); they take expectation over the finite sample $d^D$ while we assume exact expectation over $\mu$ (from which $D$ can be sampled; see also Appendix L for related discussions on generalization errors); finally, they derive the expression using fully expressive function classes $\mathcal{S}\times\mathcal{A}\to\mathbb{R}$, where our derivation always uses restricted function classes $\mathcal{Q}$ and $\mathcal{W}$.

Other than the above items, the only remaining difference is in the normalization convention: we define $J(\pi)$ and $d^\pi$ (and hence $w_{\pi/\mu}$ and $w$) all in a unnormalized manner, whereas they take the normalized versions, which is why the expression still differs by a factor of $(1-\gamma)$.

## E  Comparison to Naïve Intervals

We discuss in further details here why our upper and lower bounds are tighter than the naïve ones from previous works (see Table 1 and Remark 3). We compare $\mathrm{UB}_q$ and $\mathrm{UB}_q'$ as an example, and the situation for the other pairs of bounds are similar.

Recall that the upper bound derived from MQL [1] is

$$\mathrm{UB}_q' = \hat{q}(s_0,\pi) + \inf_{q\in\mathcal{Q}} \sup_{w\in\mathcal{W}} \left|\mathbb{E}_w[L_q(w,q)]\right|, \tag{14}$$

where $L_q(w,q) := \mathbb{E}_w[r + \gamma q(s',\pi) - q(s,a)]$, and $\hat{q} = \arg\min_{q\in\mathcal{Q}} \sup_{w\in\mathcal{W}} \left|L_q(w,q)\right|$. In comparison, our bound is

$$\mathrm{UB}_q = \inf_{q\in\mathcal{Q}} \sup_{w\in\mathcal{W}} \left(q(s_0,\pi) + L_q(w,q)\right). \tag{15}$$

Both bounds are valid upper bounds with realizable $\mathcal{C}(\mathcal{Q})$. Below we show that our upper bound is never higher than its naïve counterpart (this result does not require realizable $\mathcal{Q}$).

**Proposition 13.** $\mathrm{UB}_q \leq \mathrm{UB}_q'$.

*Proof.*

$$\mathrm{UB}_q = \inf_{q\in\mathcal{Q}} \left(q(s_0,\pi) + \sup_{w\in\mathcal{W}} L_q(w,q)\right)$$

$$\leq \hat{q}(s_0,\pi) + \sup_{w\in\mathcal{W}} L_q(w,\hat{q}) \tag{16}$$

$$\leq \hat{q}(s_0,\pi) + \sup_{w\in\mathcal{W}} |L_q(w,\hat{q})| \tag{17}$$

$$= \hat{q}(s_0,\pi) + \inf_{q\in\mathcal{Q}} \sup_{w\in\mathcal{W}} |L_q(w,q)| = \mathrm{UB}_q'. \qquad \square$$

**Remark 5.** As we can see, the tightness of $\mathrm{UB_q}$ comes from two sources: (1) that we perform a unified optimization and put $q(s_0, \pi)$ inside $\inf_{q \in \mathcal{Q}}$ (reflected in Eq.(16)), and (2) that we do not need the absolute value in our objective (reflected in Eq.(17)). On the other hand, if $\mathcal{W}$ is symmetric—that is, $-w \in \mathcal{W}, \forall w \in \mathcal{W}$—then we only enjoy the first kind of tightness.[11]

**Remark 6.** $\mathrm{UB_q} = \mathrm{UB_q}'$ requires $\sup_{w \in \mathcal{W}} L_{\mathrm{q}}(w, \hat{q}) = \sup_{w \in \mathcal{W}} |L_{\mathrm{q}}(w, \hat{q})|$ as a necessary condition. Similarly, one can show that $\mathrm{LB_q} = \mathrm{LB_q}'$ requires $-\inf_{w \in \mathcal{W}} L_{\mathrm{q}}(w, \hat{q}) = \sup_{w \in \mathcal{W}} |L_{\mathrm{q}}(w, \hat{q})|$. Therefore, as long as

$$- \sup_{q \in \mathcal{Q}} \inf_{w \in \mathcal{W}} L_{\mathrm{q}}(w, q) \neq \inf_{q \in \mathcal{Q}} \sup_{w \in \mathcal{W}} L_{\mathrm{q}}(w, q),$$

at least one side of our interval will be strictly tighter than before.

# F   Regularization

Here we show how to introduce regularization into our intervals in a way similar to [20].

**Derivation using Fenchel–Legendre Transformation**  We exemplify how to adapt the derivation in Sec. 4.1 to obtain the regularized interval in Sec. 4.1; the adaptation of Sec. 4.2 is similar which we leave to the readers. Our derivation uses Fenchel-Legendre transformation in a way similar to [20, 21]: we assume that $f : \mathbb{R} \to \mathbb{R}$ has a convex conjugate $f^*$ (i.e., $f^*(x^*) = \sup_x \{x \cdot x^* - f(x)\}$) that satisfies $f^*(0) = 0$. Below we show that by inserting such an $f^*$ into the derivation, we can obtain an interval that uses $f$ as the regularization function.

$$J(\pi) = Q^\pi(s_0, \pi) + \mathbb{E}_{s,a \sim \mu}\Big[ f^* \Big( \mathbb{E}_{r,s'|s,a}\Big[ r + \gamma Q^\pi(s', \pi) - Q^\pi(s, a) \Big] \Big) \Big]$$

$$(\mathcal{T}Q^\pi = Q^\pi \text{ and } f^*(0) = 0)$$

$$= Q^\pi(s_0, \pi) + \mathbb{E}_{s,a \sim \mu}\Big[ \sup_x \Big( x \cdot \mathbb{E}_{r,s'|s,a}\Big[ r + \gamma Q^\pi(s', \pi) - Q^\pi(s, a) \Big] - f(x) \Big) \Big]$$

$$\geq Q^\pi(s_0, \pi) + \mathbb{E}_{s,a \sim \mu}\Big[ w(s, a) \cdot \mathbb{E}_{r,s'|s,a}\Big[ r + \gamma Q^\pi(s', \pi) - Q^\pi(s, a) \Big] - f(w(s, a)) \Big]$$

$$(\text{Replace } x \text{ for each } (s, a) \text{ by } w(s, a) \text{ using an arbitrary } w \in \mathcal{W})$$

$$= Q^\pi(s_0, \pi) + \mathbb{E}_\mu\Big[ w(s, a) \cdot \Big( r + \gamma Q^\pi(s', \pi) - Q^\pi(s, a) \Big) - f(w(s, a)) \Big].$$

Although $f$ may be nonlinear, it is only applied to $w$ and the entire expression is still affine in $Q^\pi$, so we can replace $Q^\pi$ with $\inf_{q \in \mathcal{Q}}$ as before:

$$J(\pi) \geq \inf_{q \in \mathcal{Q}} q(s_0, \pi) + \mathbb{E}_\mu\Big[ w(s, a) \cdot \Big( r + \gamma Q^\pi(s', \pi) - Q^\pi(s, a) \Big) - f(w(s, a)) \Big].$$

Since it holds for arbitrary $w$, we take $\sup_{w \in \mathcal{W}}$ and obtain the lower bound:

$$J(\pi) \geq \sup_{w \in \mathcal{W}} \left( \inf_{q \in \mathcal{Q}} L(w, q) - \mathbb{E}_\mu[f(w(s, a))] \right). \tag{18}$$

Similarly, the upper bound (under $Q^\pi \in \mathcal{C}(\mathcal{Q})$) is

$$J(\pi) \leq \inf_{w \in \mathcal{W}} \left( \sup_{q \in \mathcal{Q}} L(w, q) + \mathbb{E}_\mu[f(w(s, a))] \right). \tag{19}$$

**Properties of the Regularized Interval**  From the above derivation, we see that the regularized interval is valid when $Q^\pi \in \mathcal{C}(\mathcal{Q})$, which is the same condition needed for the validity of the unregularized interval in Sec. 4.1. One may naturally wonder what their relationship is, and the answer is very simple: adding any nontrivial regularization loosens the interval.

To see this, first notice that Eq.(18) and (19) differ from $\mathrm{LB_w}$ and $\mathrm{UB_w}$ by a term $\mathbb{E}_\mu[f(w(s, a))]$: such a term is subtracted from the lower bound and added to the upper bound. Now recall that our derivation crucially relies on $f^*(0) = 0$. A direct consequence is that $f$ must be non-negative, as $f^*(0) = \inf_x f(x)$. Therefore, the regularization increases the upper bound and decreases the lower bound, which makes the interval looser. The tightest bound is obtained without any regularization, i.e., $\mathrm{LB_w}$ and $\mathrm{UB_w}$, which corresponds to $f \equiv 0$, whose convex conjugate is

$$f^*(x^*) = \begin{cases} 0, & x = 0; \\ +\infty, & x \neq 0. \end{cases}$$

Figure 5: Example training curves averaged over 10 seeds. Error bars show twice the standard errors. The curves have been smoothed with a window size corresponding to 20 $q$-$w$ alternations. **Top**: The difference between behavior policy and target policy is relatively small, and $\mathrm{UB_q} > \mathrm{LB_q}$. The interval $[\mathrm{LB_q}, \mathrm{UB_q}]$ is significantly tighter than $[\mathrm{LB_q}', \mathrm{UB_q}']$. **Bottom**: Target policy is significantly different from the behavior policy, and realizing the importance weight is challenging. In this case, a reversed interval is observed, i.e. $\mathrm{UB_q} < \mathrm{LB_q}$.

## G   OPE Experiments

### G.1   Environment and Behavior & Target Policies

We conduct experiments in the CartPole environment with $\gamma = 0.99$. Following Uehara et al. [1], we modify the reward function and add small Gaussian noise to transition dynamics to make OPE more challenging in this environment.[12] To generate the behavior and the target policies, we apply softmax on a near-optimal $Q$-function trained via the open source code[13] of DQN with an adjustable temperature parameter $\tau$:

$$\pi(a|s) \propto \exp(\frac{Q(s,a)}{\tau}). \tag{20}$$

The behavior policy $\pi_b$ is chosen as $\tau = 1.0$, and we use other values of $\tau$ for target policies. To collect the dataset, we truncate the generated trajectories at the 1000-th time step. For those terminated within 1000 steps, we pad the rest of the trajectories with the terminal states. We treat $\mu = d^\pi$ and approximate such a data distribution by weighting each data point $(s, a, r, s')$ with a weight $\gamma^t$, where $t$ is the time step $s$ is observed. All experiments generate datasets of 200 trajectories except for the interval length comparison (Fig. 3), where the sample size is indicated on the x-axis. We report average results over 10 seeds in all the OPE experiments, and show twice the standard errors as error bars which correspond to 95% confidence intervals.

### G.2   Details of the Algorithms

We compare $\mathrm{UB_q}$ and $\mathrm{LB_q}$ to $\mathrm{UB_q}'$ and $\mathrm{LB_q}'$ in Table 1. We use Multilayer Perceptron (MLP) to construct $\mathcal{Q}$ and $\mathcal{W}$. The detailed specification of $\mathcal{Q}$ will be given later. For $\mathcal{W}$, the ideal choice denoted by $\mathcal{W}^\alpha$ is defined as:

$$\mathcal{W}^\alpha = \Big\{ w(\cdot, \cdot) = \frac{\alpha |f(\cdot, \cdot)|}{\mathbb{E}_\mu[|f|]} \Big| f \in \mathrm{MLP} \Big\}.$$

Note that normalizing with $\mathbb{E}_\mu[|f|]$ allows us to directly control the expectation of any $w \in \mathcal{W}$ to be $\alpha$, and we use $\alpha = 25/(1 - \gamma)$ throughout the OPE experiments. However, directly optimizing over $\mathcal{W}^\alpha$ is quite unstable, and we consider the following relaxation of our upper and lower bounds:

fixing any $q \in \mathcal{Q}$, for any $w \in \mathcal{W}^\alpha$, we relax the $\mathbb{E}_w[r + \gamma q(s', \pi) - q(s, a)]$ term in $\mathrm{UB_q}$ as

$$
\begin{aligned}
&\mathbb{E}_w\Big[r + \gamma q(s', \pi) - q(s, a)\Big] \\
=&\mathbb{E}_\mu\Big[(r + \gamma q(s', \pi) - q(s, a))\frac{\alpha|f|}{\mathbb{E}_\mu[|f|]}\Big] \\
\leq&\mathbb{E}_\mu\Big[(r + \gamma q(s', \pi) - q(s, a))\frac{\alpha|f|\mathbb{I}[r + \gamma q(s', \pi) - q(s, a) > 0]}{\mathbb{E}_\mu[|f|]}\Big],
\end{aligned}
\tag{21}
$$

where $\mathbb{I}[\cdot] = 1$ if the predicate is true, and $0$ otherwise. So essentially we turn each $f$ into a weighting function $w$ that evaluates to $\frac{\alpha|f|\mathbb{I}[r+\gamma q(s',\pi)-q(s,a)>0]}{\mathbb{E}_\mu[|f|]}$ on each data point $(s, a, r, s')$. Such a relaxation helps stabilize training and results in a looser upper bound compared to $\mathrm{UB_q}$, which is still valid as long as $\mathrm{UB_q}$ is a valid upper bound. We similarly relax the lower bound by replacing $\mathbb{I}[r + \gamma q(s', \pi) - q(s, a) > 0]$ with $\mathbb{I}[r + \gamma q(s', \pi) - q(s, a) < 0]$.

In addition, although optimizing MQL loss with $\mathcal{W}^\alpha$ can converge, we find that using the same relaxation can further stabilize training and lead to better results. Therefore, we adopt this trick in the calculation of $\mathrm{UB_q}'$ and $\mathrm{LB_q}'$ as well.

We use a $32 \times 32$ MLP (with tanh activation) to parameterize $q$ (except in Fig 1 where the architecture is indicated on the x-axis) and use a one-hidden-layer MLP with 32 units for $f$ (which produces $w$). Moreover, we clip $q$ in the interval $[\frac{R_{\min}}{1-\gamma} - \delta, \frac{R_{\max}}{1-\gamma}]$, where $R_{\min} = 0.0$, $R_{\max} = 3.0$, $\delta = 20$. We use stochastic gradient descent ascent (SGDA) with minimatches for optimization, and alternate between $q$ and $w$ every 500 and 50 iterations, respectively. The learning rates are both fixed as $0.005$, and each minibatch consists of $500$ transition tuples. The normalization factor in the relaxed objective is approximated on the minibatch. During the $q$-optimization phases, the weights $w$ (which depends on $q$ through the indicators) is treated as a constant and does not contribute to the gradients, but is re-computed every time $q$ is updated. See example training curves in Fig. 5.

### G.3 Bootstrapping Intervals

To account for statistical errors in our interval, we sample the dataset with replacement to generate 20 bootstrapped datasets with the same size as the original dataset. Then, we run our algorithms on those new datasets and record the upper/lower bounds. We pick the $k$-th largest (smallest) upper (lower) bound as the final upper (lower) bound, and for figures shown here we use $k = 1$. We report interval lengths and validity ratios averaged over 10 runs.

### G.4 On Comparison to MQL

The comparison in Fig. 3 should be interpreted carefully. This figure is used to demonstrate the theoretical predictions in App. E, where the two methods use the same $\mathcal{Q}$ and $\mathcal{W}$ classes. Note that our choice of $\mathcal{W}$ class in this experiment, $\mathcal{W}^\alpha$, comes with a hyperparameter $\alpha$ that adjusts the magnitude of the functions in the class, and we choose a fairly large value of $\alpha = 25/(1 - \gamma)$ for stability of optimization. (A more principled optimization approach is also an interesting future direction, which may allow us to choose a much smaller value of $\alpha$ for our intervals.) The loss of MQL, on the other hand, is homogeneous in $w \in \mathcal{W}$ hence the algorithm is *invariant* to the rescaling of $\mathcal{W}$. Therefore, the value of $\alpha$ has no effect on the training process and merely determines the length of the interval in a straightforward manner (i.e., when we change $\alpha$, MQL's interval length scales linearly with $\alpha$, and its center does not move). To this end, the $\alpha$ for MQL could have been tuned to obtain a tighter yet still valid interval, although this is difficult in practice as we do not have the groundtruth value to tune $\alpha$ against (otherwise OPE would not be necessary; see discussions in [33]). Nevertheless, we reiterate that Fig. 3 is only meant to empirically illustrate the theoretical predictions of App. E, and to compare the two methods fairly the value of $\alpha$ should be tuned for MQL in some fashion, which we do not investigate in this paper.

# H Sanity-Check Experiments

In App. G we introduced several tricks to stabilize the difficult minimax optimization problems associated with our intervals. While the tricks have worked out empirically in CartPole, we would like to further understand the legitimacy and the consequences of these tricks.

The major modification is the relaxation in Eq.(21), where we allow $w \in \mathcal{W}$ to depend not only on $(s, a)$ but also on the randomness of $(r, s')$. A potential caveat is that, if the transition or reward function are stochastic, given two tuples $(s_1, a_1, r_1, s'_1)$ and $(s_2, a_2, r_2, s'_2)$, it is possible that $w(s_1, a_1) \neq w(s_2, a_2)$ if $s_1 = s_2, a_1 = a_2$ but $r_1 \neq r_2$ or $s'_1 \neq s'_2$. In comparison, without the relaxation in Eq.(21), we shall always have $w(s_1, a_1) = w(s_2, a_2)$. While we may hardly find two identical state-action pairs in continuous tasks, the issue still exists if we observe states that are considered close to each other by the function approximator $\mathcal{W}$. Therefore, we hypothesize that this relaxation may make the interval loose when the environment is highly stochastic, and an ideal way to test this hypothesis is to conduct an experiment in a tabular environment because (1) it is easy to find tabular environments with sufficient stochasticity, and (2) we can afford to implement a more faithful version of our algorithm in the tabular setting, which could be unstable and even diverge in the function-approximation setting.

**Experiment Setup** Given the above considerations, we conduct additional experiments in Taxi, a tabular environment with 2000 states and 6 actions. We parameterize $Q$ with $|\mathcal{S}| \times |\mathcal{A}|$ bounded variables, i.e., $\mathcal{Q} = \left[ \frac{R_{\min}}{1-\gamma}, \frac{R_{\max}}{1-\gamma} \right]^{\mathcal{S} \times \mathcal{A}}$. As for $\mathcal{W}$, we consider three different function classes:

$$
\begin{aligned}
\mathcal{W}_1^\alpha = & \left\{ w(s,a) = \frac{\alpha f(s,a) \mathbb{I}[r + \gamma Q(s', \pi) - Q(s,a) > 0]}{\mathbb{E}_{(\tilde{s}, \tilde{a}, \tilde{r}, \tilde{s}') \sim \mu}[f(\tilde{s}, \tilde{a})]}, \forall (s,a) \in \mathcal{S} \times \mathcal{A} \right\} \\
& \cup \left\{ w(s,a) = \frac{\alpha f(s,a) \mathbb{I}[r + \gamma Q(s', \pi) - Q(s,a) < 0]}{\mathbb{E}_{(\tilde{s}, \tilde{a}, \tilde{r}, \tilde{s}') \sim \mu}[f(\tilde{s}, \tilde{a})]}, \forall (s,a) \in \mathcal{S} \times \mathcal{A} \right\} \\
\mathcal{W}_2^\alpha = & \left\{ w(s,a) = \frac{\alpha f(s,a) \mathbb{I}[\mathbb{E}_{r,s'|s,a}[r + \gamma Q(s', \pi) - Q(s,a)] > 0]}{\mathbb{E}_{(\tilde{s}, \tilde{a}) \sim \mu}[f(\tilde{s}, \tilde{a})]}, \forall (s,a) \in \mathcal{S} \times \mathcal{A} \right\} \\
& \cup \left\{ w(s,a) = \frac{\alpha f(s,a) \mathbb{I}[\mathbb{E}_{r,s'|s,a}[r + \gamma Q(s', \pi) - Q(s,a)] < 0]}{\mathbb{E}_{(\tilde{s}, \tilde{a}) \sim \mu}[f(\tilde{s}, \tilde{a})]}, \forall (s,a) \in \mathcal{S} \times \mathcal{A} \right\} \\
\mathcal{W}_3^\alpha = & \left\{ w(s,a) = \frac{\alpha f(s,a)}{\mathbb{E}_{(\tilde{s}, \tilde{a}) \sim \mu}[f(\tilde{s}, \tilde{a})]}, \forall (s,a) \in \mathcal{S} \times \mathcal{A} \right\}
\end{aligned}
$$

where $f$ can take independent values in different $(s, a)$ (i.e., we use a tabular representation for $f$). We only constrain $f$ to be positive but do not clip its value; during the training processes its value is usually bounded.

As we can see, $\mathcal{W}_1^\alpha$ precisely corresponds to the relaxation in Eq.(21), where we replaced the MLP $f$ with a tabular $f$. We compare it to two alternatives: $\mathcal{W}_2^\alpha$ and $\mathcal{W}_3^\alpha$: in $\mathcal{W}_2^\alpha$, we first integrate out the randomness in $\mathbb{E}_{r,s'|s,a}$ so that $w$ does not depend on $(r, s')$, but still keep the modification related to the use of indicator function $\mathbb{I}[\ldots > 0]$ that avoids negativity. This indicator trick is further removed in $\mathcal{W}_3^\alpha$, where we have a completely faithful implementation of the original algorithm. When optimizing with $\mathcal{W}_1^\alpha$ and $\mathcal{W}_2^\alpha$, we adopt SGDA and alternate between optimizing $Q$ and $w$ every 250 iterations. The learning rate for both $Q$ and $w$ are 5e-3, and the batch size is 500. When optimizing with $\mathcal{W}_3^\alpha$, we found that SGDA can lead to instability, and we instead update $Q$ and $w$ synchronously at each training iteration. (As a side note, synchronous updates fail in CartPole experiments.) We use the whole dataset in each iteration, and the learning rate is fixed to 5e-3.

The results are showed in Figure 6. As we can see, the dependence of $w$ on $(r, s')$ indeed result in a looser bound asymptotically. Moreover, the use of the indicator trick does not make the interval loose in this case, while it converges more slowly than synchronously updating $Q$ and $w$ with $\mathcal{W}_3^\alpha$. The looseness of $\mathcal{W}_1^\alpha$ suggests that our optimization strategy in Appendix G is not general enough, and it will be important to investigate more principled optimization strategies that directly work with the original optimization problem without the relaxation in Eq.(21).

Figure 6: Comparison between experiments on Taxi with different function classes. Average over 3 seeds. **Red Curves**: Upper/Lowers bound using $\mathcal{W}_1^\alpha$. **Blue Curves**: Upper/Lower bounds using $\mathcal{W}_2^\alpha$. **Green Curves**: Upper/Lower bounds using $\mathcal{W}_3^\alpha$.

# I  Rmax / Rmin

Here we review the concept of Rmax and Rmin in detail, and provide the proof of Proposition 9. We first recall the setup of Sec. 5.1:

**Setup**  Consider an MDP with finite and discrete state and action spaces. Let $\tilde{\mathcal{S}} \subset \mathcal{S}$ be a subset of the state space, and $s_0 \in \tilde{\mathcal{S}}$. Suppose the dataset contains $n$ transition samples from each $s \in \tilde{\mathcal{S}}, a \in \mathcal{A}$ with $n \to \infty$ (i.e., the transitions and rewards in $\tilde{\mathcal{S}}$ are fully known), and 0 samples from $s \notin \tilde{\mathcal{S}}$.

**Rmax and Rmin**  In such a simplified setting, effective exploration can be achieved by (the episodic version of) the well-known Rmax algorithm [25, 26], which computes the optimal policy of the Rmax-MDP $M_{\max}$, defined as $(\mathcal{S}, \mathcal{A}, P_{M_{\max}}, R_{M_{\min}}, \gamma, s_0)$, where[14]

$$P_{M_{\max}}(s,a) = \begin{cases} P(s,a), & \text{if } s \in \tilde{\mathcal{S}} \\ \mathbb{I}[s' = s], & \text{if } s \notin \tilde{\mathcal{S}} \end{cases}, \qquad R_{M_{\max}}(s,a) = \begin{cases} R(s,a), & \text{if } s \in \tilde{\mathcal{S}} \\ R_{\max}, & \text{if } s \notin \tilde{\mathcal{S}} \end{cases}.$$

In words, $M_{\max}$ is the same as the true MDP $M$ on states with sufficient data, and the remaining "unknown" states are assumed to have self-loops with $R_{\max}$ immediate rewards, making them appealing to visit and thus encouraging exploration. The theoretical guarantee for the policy computed by Rmax is an optimal-or-explore statement [see e.g., 34], that either the optimal policy of $M_{\max}$ is near-optimal in true $M$, or its occupancy measure must visit states outside $\tilde{\mathcal{S}}$, with a mass proportional to its suboptimality. Similarly, when the goal is to exploit, that is, to output a policy with the best worst-case guarantee, one simply needs to change the $R_{\max}$ in the Rmax-MDP to the minimally possible reward value (which is 0 in our setting), and we call this MDP $M_{\min}$.

**Remark 7.**  In the simplified setting above, the states outside $\tilde{\mathcal{S}}$ receive no data at all. In the more general case where $\tilde{\mathcal{S}}$ is still under-explored but every state receives any least 1 data point, it is not difficult to show that both MUB-PO and MLB-PO reduce to the certainty-equivalent solution, and can overfit to the poor estimation of transitions and rewards in states with few data points. Such a degenerate behavior may be prevented by regularizing $w$ and prohibiting any highly spiked $w$ (e.g., regularizing with $\mathbb{E}_\mu[w^2]$, which measures the effective sample size of importance weighted estimate induced by $w$), or by bootstrapping and taking data randomness into consideration during policy optimization.

***Proof of Proposition 9***.  Let $\mathcal{M}$ be the space of all MDPs that are consistent with the given dataset. Let $M_{\max}$ and $M_{\min}$ be the Rmax and Rmin MDPs, resp., and it is clear that $M_{\max}, M_{\min} \in \mathcal{M}$.

**MLB-PO Reduces to Rmin**  To show that MLB-PO reduces to Rmin, it suffices to prove that for every $\pi : \mathcal{S} \to \mathcal{A}$, $\text{LB}_w^\pi = J_{M_{\min}}(\pi)$. Since $\mathcal{Q}$ is the tabular function space and always

realizable (regardless of the true MDP $M$), $\mathrm{LB}_\mathrm{w}^\pi$ is a valid lower bound, i.e., $\mathrm{LB}_\mathrm{w}^\pi \le J(\pi)$. Since all MDPs in $\mathcal{M}$ *could* be the true $M$, it must hold that $\mathrm{LB}_\mathrm{w}^\pi \le J_{M_\mathrm{min}}(\pi)$. It suffices to show that $\mathrm{LB}_\mathrm{w}^\pi \ge J_{M_\mathrm{min}}(\pi)$.

Recall that

$$\mathrm{LB}_\mathrm{w}^\pi = \sup_{w \in \mathcal{W}} \inf_{q \in \mathcal{Q}} \{q(s_0, \pi) + \mathbb{E}_w[r + \gamma q(s', \pi) - q(s, a)]\}.$$

Since $\mathcal{Q}$ is the unrestricted tabular function space, we may represent $q \in \mathcal{Q}$ as a $|\mathcal{S} \times \mathcal{A}|$ vector where each coordinate $q(s, a)$ can take values between $[0, R_\mathrm{max}/(1 - \gamma)]$ independently. Now note that for any $s' \notin \tilde{\mathcal{S}}$, $a' \in \mathcal{A}$, $q(s', a')$ as a decision variable may only appear as the $\gamma q(s', \pi)$ term in the objective and can never appear as $q(s_0, \pi)$ or $-q(s, a)$. Since $\mathcal{W}$ only contains non-negative functions, $L(w, q)$ is non-decreasing in $q(s', \pi)$, and $\inf_{q \in \mathcal{Q}}$ can always be attained when $q(s', a') = 0, \forall s' \notin \tilde{\mathcal{S}}, a' \in \mathcal{A}$. For any $w \in \mathcal{W}$, let $q_w$ denote such a $q \in \mathcal{Q}$ that achieves the inner infimum, and

$$\mathrm{LB}_\mathrm{w}^\pi = \sup_{w \in \mathcal{W}} \{q_w(s_0, \pi) + \mathbb{E}_w[r + \gamma q_w(s', \pi) - q_w(s, a)]\}.$$

Next, consider the discounted occupancy of $\pi$ in $M_\mathrm{min}$, denoted as $d_{M_\mathrm{min}}^\pi$. Define

$$w_0(s, a) := \mathbb{I}[s \in \tilde{\mathcal{S}}] \frac{d_{M_\mathrm{min}}^\pi(s, a)}{1/|\tilde{\mathcal{S}} \times \mathcal{A}|}.$$

Let $q_0 = q_{w_0}$. Now, since $w_0 \in \mathcal{W}$,

$$\mathrm{LB}_\mathrm{w}^\pi \ge q_0(s_0, \pi) + \mathbb{E}_{w_0}[r + \gamma q_0(s', \pi) - q_0(s, a)]$$

$$= q_0(s_0, \pi) + \frac{1}{|\tilde{\mathcal{S}} \times \mathcal{A}|} \sum_{s \in \tilde{\mathcal{S}}, a \in \mathcal{A}} w_0(s, a)(R(s, a) + \gamma \mathbb{E}_{s' \sim P(s, a)}[q_0(s', \pi)] - q_0(s, a))$$

$$= q_0(s_0, \pi) + \sum_{s \in \tilde{\mathcal{S}}, a \in \mathcal{A}} d_{M_\mathrm{min}}^\pi(s, a)(R(s, a) + \gamma \mathbb{E}_{s' \sim P(s, a)}[q_0(s', \pi)] - q_0(s, a))$$

$$= \sum_{s \in \tilde{\mathcal{S}}, a \in \mathcal{A}} d_{M_\mathrm{min}}^\pi(s, a)R(s, a) + \sum_{s \in \mathcal{S}, a \in \mathcal{A}} \nu(s, a)q_0(s, a),$$

where

$$\nu(s, a) := \mathbb{I}[s = s_0, a = \pi(s_0)] + \gamma \sum_{s' \in \tilde{\mathcal{S}}, a' \in \mathcal{A}} d_{M_\mathrm{min}}^\pi(s', a')P(s|s', a') - \mathbb{I}[s \in \tilde{\mathcal{S}}]d_{M_\mathrm{min}}^\pi(s, a).$$

Note that

$$\sum_{s \in \tilde{\mathcal{S}}, a \in \mathcal{A}} d_{M_\mathrm{min}}^\pi(s, a)R(s, a) = \sum_{s \in \tilde{\mathcal{S}}, a \in \mathcal{A}} d_{M_\mathrm{min}}^\pi(s, a)R(s, a) + \sum_{s \notin \tilde{\mathcal{S}}, a \in \mathcal{A}} d_{M_\mathrm{min}}^\pi(s, a) \cdot 0 = J_{M_\mathrm{min}}(\pi),$$

so it suffices to show that $\sum_{s \in \mathcal{S}, a \in \mathcal{A}} \nu(s, a)q_0(s, a) = 0$, which we establish in the rest of this proof.

Recall that $q_0(s, a) = 0$ for any $s \notin \tilde{\mathcal{S}}$. So $\sum_{s \in \mathcal{S}, a \in \mathcal{A}} \nu(s, a)q_0(s, a) = \sum_{s \in \mathcal{S}, a \in \mathcal{A}} \nu'(s, a)q_0(s, a)$, where

$$\nu'(s, a) := \mathbb{I}[s = s_0, a = \pi(s_0)] + \gamma \sum_{s' \in \mathcal{S}, a' \in \mathcal{A}} d_{M_\mathrm{min}}^\pi(s', a')P_{M_\mathrm{min}}(s|s', a') - d_{M_\mathrm{min}}^\pi(s, a).$$

This is because $\nu$ and $\nu'$ exactly agree on the value in $s \in \tilde{\mathcal{S}}$, and only differ on $s \notin \tilde{\mathcal{S}}$. To see this, consider any $s \in \tilde{\mathcal{S}}, a \in \mathcal{A}$. $\nu$ and $\nu'$ agree on the first and the last terms. They also agree on the second term for the summation variables $s' \in \tilde{\mathcal{S}}, a' \in \mathcal{A}$, as $P(s|s', a') = P_{M_\mathrm{min}}(s|s', a')$ (the MDP $M_\mathrm{min}$ has the true transition probabilities on states with sufficient data). So the only difference is

$$\gamma \sum_{s' \notin \tilde{\mathcal{S}}, a' \in \mathcal{A}} d_{M_\mathrm{min}}^\pi(s', a')P_{M_\mathrm{min}}(s|s', a').$$

However, this term must be zero, because for any $s \in \tilde{\mathcal{S}}, s' \notin \tilde{\mathcal{S}}$, $P_{M_\mathrm{min}}(s|s', a') = 0$, as the construction of $M_\mathrm{min}$ guarantees that no states outside $\tilde{\mathcal{S}}$ will ever transition back to $\tilde{\mathcal{S}}$. Now that we conclude $\nu(s, a)q_0(s, a) = \nu'(s, a)q_0(s, a)$, the fact that it is zero is obvious: $\nu' \equiv 0$, which follows directly from the Bellman equation for discounted occupancy in MDP $M_\mathrm{min}$. This completes the proof of $\mathrm{LB}_\mathrm{w}^\pi = J_{M_\mathrm{min}}(\pi)$.

**MUB-PO Reduces to Rmax** The proof for $\text{UB}_w^\pi = J_{M_{\max}}(\pi)$ is similar. Using the same argument as above, we know that $\text{UB}_w^\pi \geq J_{M_{\max}}(\pi)$ and it suffices to show that $\text{UB}_w^\pi \leq J_{M_{\max}}(\pi)$.

Recall that

$$\text{UB}_w^\pi = \inf_{w \in \mathcal{W}} \sup_{q \in \mathcal{Q}} \left\{ q(s_0, \pi) + \mathbb{E}_w[r + \gamma q(s', \pi) - q(s, a)] \right\}.$$

Next, we consider the discounted occupancy of $\pi$ in $M_{\max}$, denoted as $d_{M_{\max}}^\pi$. Define $w_0$ as (we recycle the symbol from the previous part of the proof)

$$w_0(s, a) := \mathbb{I}[s \in \tilde{\mathcal{S}}] \frac{d_{M_{\max}}^\pi(s, a)}{1/|\tilde{\mathcal{S}} \times \mathcal{A}|}.$$

and $q_0$ as

$$q_0 := \arg\max_{q \in \mathcal{Q}} \left\{ q(s_0, \pi) + \mathbb{E}_{w_0}[r + \gamma q(s', \pi) - q(s, a)] \right\}.$$

With the same argument as the Rmin case, we can always have $q_0(s, a) = \frac{R_{\max}}{1-\gamma}, \forall s \notin \tilde{\mathcal{S}}, a \in \mathcal{A}$.

Since $w_0 \in \mathcal{W}$,

$$\text{UB}_w^\pi \leq q_0(s_0, \pi) + \mathbb{E}_{w_0}[r + \gamma q_0(s', \pi) - q_0(s, a)]$$

$$= q_0(s_0, \pi) + \frac{1}{|\tilde{\mathcal{S}} \times \mathcal{A}|} \sum_{s \in \tilde{\mathcal{S}}, a \in \mathcal{A}} w_0(s, a)(R(s, a) + \gamma \mathbb{E}_{s' \sim P(s,a)}[q_0(s', \pi)] - q_0(s, a))$$

$$= q_0(s_0, \pi) + \sum_{s \in \tilde{\mathcal{S}}, a \in \mathcal{A}} d_{M_{\max}}^\pi(s, a)(R(s, a) + \gamma \mathbb{E}_{s' \sim P(s,a)}[q_0(s', \pi)] - q_0(s, a))$$

$$= q_0(s_0, \pi) + J_{M_{\max}}(\pi) - \sum_{s \notin \tilde{\mathcal{S}}, a \in \mathcal{A}} d_{M_{\max}}^\pi(s, a) R_{\max}$$

$$+ \sum_{s \in \tilde{\mathcal{S}}, a \in \mathcal{A}} d_{M_{\max}}^\pi(s, a)(\gamma \mathbb{E}_{s' \sim P(s,a)}[q_0(s', \pi)] - q_0(s, a)).$$

The last step follows from the fact that in $M_{\max}$, all $s \notin \tilde{\mathcal{S}}$ are absorbing states with $R_{\max}$ rewards. Now that we have extracted out $J_{M_{\max}}(\pi)$, it suffices to show that the remaining terms cancel.

To show the cancellation, we will use two properties of the $M_{\max}$ construction: (1) For any $s \notin \tilde{\mathcal{S}}$, $q_0(s, a) - \gamma \mathbb{E}_{s' \sim P_{M_{\max}}(s,a)}[q_0(s', \pi)] = \frac{R_{\max}}{1-\gamma} - \frac{\gamma R_{\max}}{1-\gamma} = R_{\max}$ (because $P_{M_{\max}}(s, a)$ is a point mass on $s$), and (2) for any $s \in \tilde{\mathcal{S}}$, $P_{M_{\max}}(s, a) = P(s, a)$. Using these, we may rewrite the remaining terms as

$$q_0(s_0, \pi) - \sum_{s \notin \tilde{\mathcal{S}}, a \in \mathcal{A}} d_{M_{\max}}^\pi(s, a) R_{\max} + \sum_{s \in \tilde{\mathcal{S}}, a \in \mathcal{A}} d_{M_{\max}}^\pi(s, a)(\gamma \mathbb{E}_{s' \sim P(s,a)}[q_0(s', \pi)] - q_0(s, a))$$

$$= q_0(s_0, \pi) + \sum_{s \in \mathcal{S}, a \in \mathcal{A}} d_{M_{\max}}^\pi(s, a)(\gamma \mathbb{E}_{s' \sim P_{M_{\max}}(s,a)}[q_0(s', \pi)] - q_0(s, a)).$$

This is 0 due to the Bellman equation for occupancy $d_{M_{\max}}^\pi$ in the MDP $M_{\max}$ (which we have also used in Lemma 4). $\qquad\square$

## J  Additional Proofs, Results, and Discussions of Section 5

### J.1  Proof of Proosition 10

It suffices to show that $J(\hat{\pi}) \geq \text{LB}_w^{\hat{\pi}}$, which follows directly from Theorem 2: $\text{LB}_w^\pi$ is a valid lower bound for any $\pi \in \Pi$ due to the realizability of $\mathcal{Q}$. The corollary holds because for $\pi$ with $w_{\pi/\mu}$ realized by $\mathcal{W}$, Theorems 2 and 3 guarantee that $J(\pi) = \text{LB}_w^\pi$. $\qquad\square$

## J.2 Full Version of Proposition 11 and Proof

**Proposition 14** (Full version of Proposition 11). *Let $\Pi$ be a policy class. Let $\hat{\pi} = \arg\max_{\pi \in \Pi} \mathrm{UB}_{\mathrm{w}}^{\pi}$. Then, for any $w \in \mathcal{W}$,*

$$\mathrm{IPM}(w \cdot \mu, d^{\hat{\pi}}; \mathcal{F}) \geq \max_{\pi \in \Pi: Q^{\pi} \in \mathcal{C}(\mathcal{Q})} J(\pi) - J(\hat{\pi}),$$

*where*

- $(w \cdot \mu)(s, a) := w(s, a) \cdot \mu(s, a)$,
- $\mathrm{IPM}(\nu_1, \nu_2; \mathcal{F}) := \sup_{f \in \mathcal{F}} |\mathbb{E}_{\nu_1}[f] - \mathbb{E}_{\nu_2}[f]|$,
- $\mathcal{F} := \{\mathcal{T}^{\pi} q - q : q \in \mathcal{Q}, \pi \in \Pi\}$.

*As a corollary, if we further assume $\|q\|_{\infty} \leq R_{\max}/(1-\gamma), \forall q \in \mathcal{Q}$,*

$$\|w \cdot \mu - d^{\hat{\pi}}\|_1 \geq \frac{(1-\gamma)\left(\max_{\pi \in \Pi: Q^{\pi} \in \mathcal{C}(\mathcal{Q})} J(\pi) - J(\hat{\pi})\right)}{2R_{\max}}.$$

To interpret the result, recall that $w \in \mathcal{W}$ is supposed to model an importance weight function that coverts the data distribution to the occupancy measure of some policy, e.g., $w_{\pi/\mu} \cdot \mu = d^{\pi}$. The proposition states that either $\hat{\pi}$ is near-optimal, or it will induce an occupancy measure that cannot be accurately modeled by *any* importance weights in $\mathcal{W}$ when applied on the current data distribution $\mu$. The distance[15] between the two distributions $d^{\pi}$ and $w \cdot \mu$ is measured by the Integral Probability Metric [35] defined w.r.t. a discriminator class $\mathcal{F}$, and can be relaxed to the looser but simpler $\ell_1$ distance. Therefore, if we have a rich $\mathcal{W}$ class that models all distributions covered by $\mu$, then $\hat{\pi}$ must visit new areas in the state-action space or it must be near-optimal.

Below we give the proof of Proposition 14, which reuses many results established in Sec. 4.

*Proof of Proposition 14.* Fixing any $\pi \in \Pi$ such that $Q^{\pi} \in \mathcal{C}(\mathcal{Q})$:

$$J(\pi) - J(\hat{\pi}) \leq \mathrm{UB}_{\mathrm{w}}^{\pi} - J(\hat{\pi}) \qquad (\mathrm{UB}_{\mathrm{w}} \text{ is valid upper bound as } Q^{\pi} \text{ realized by } \mathcal{C}(\mathcal{Q}))$$
$$\leq \mathrm{UB}_{\mathrm{w}}^{\hat{\pi}} - J(\hat{\pi}). \qquad (\hat{\pi} \text{ optimizes } \mathrm{UB}_{\mathrm{w}}^{\pi})$$

Recall that $\mathrm{UB}_{\mathrm{w}}^{\hat{\pi}} = \inf_{w \in \mathcal{W}} \sup_{q \in \mathcal{Q}} L(w, q; \hat{\pi})$, so for any $w \in \mathcal{W}$, $\mathrm{UB}_{\mathrm{w}}^{\hat{\pi}} \leq \sup_{q \in \mathcal{Q}} L(w, q; \hat{\pi})$. On the other hand, for any $q$, $J(\hat{\pi}) = L(w_{\hat{\pi}/\mu}, q)$. Now let $q_w := \arg\max_{q \in \mathcal{Q}} L(w, q; \hat{\pi})$. For any $w \in \mathcal{W}$,

$$\max_{\pi \in \Pi: Q^{\pi} \in \mathcal{Q}} J(\pi) - J(\hat{\pi}) \leq \mathrm{UB}_{\mathrm{w}}^{\hat{\pi}} - J(\hat{\pi})$$

$$\leq L(w, q_w; \hat{\pi}) - L(w_{\hat{\pi}/\mu}, q_w; \hat{\pi})$$
$$= \mathbb{E}_w[r + \gamma q_w(s', \hat{\pi}) - q_w(s, a)] - \mathbb{E}_{w_{\hat{\pi}/\mu}}[r + \gamma q_w(s', \hat{\pi}) - q_w(s, a)]$$
$$\leq |\mathbb{E}_w[\mathcal{T}^{\hat{\pi}} q_w - q_w]] - \mathbb{E}_{w_{\hat{\pi}/\mu}}[\mathcal{T}^{\hat{\pi}} q_w - q_w]|.$$

The main statement immediately follows by noticing that $\hat{\pi} \in \Pi, q_w \in \mathcal{Q}$, hence $\mathcal{T}^{\pi} q_w - q_w \in \mathcal{F}$. The $\ell_1$-distance corollary follows from relaxing IPM using Hölder's inequality for the $\ell_1$ and $\ell_{\infty}$ pair. $\qquad\square$

## J.3 Alternative Guarantee for MLB-PO

Proposition 10 shows that MLB-PO puts the heavy expressivity burden on $\mathcal{Q}$ and is agnostic against misspecified $\mathcal{W}$. When data has sufficient coverage and $\mathcal{W}$ is highly expressive, we can similarly show that optimizing $\mathrm{LB}_{\mathrm{q}}^{\pi}$ performs robust exploitation and is agnostic against misspecified $\mathcal{Q}$.

**Proposition 15** (Exploitation with expressive $\mathcal{W}$). *Let $\Pi$ be a policy class, and assume $w_{\pi/\mu} \in \mathcal{C}(\mathcal{W}) \, \forall \pi \in \Pi$. Let $\hat{\pi} = \arg\max_{\pi \in \Pi} \mathrm{LB}_{\mathrm{q}}^{\pi}$. Then, for any $\pi \in \Pi$,*

$$J(\hat{\pi}) \geq \mathrm{LB}_{\mathrm{q}}^{\pi}.$$

*As a corollary, for any $\pi$ such that $Q^{\pi} \in \mathcal{Q}$, we have $J(\hat{\pi}) \geq J(\pi)$, that is, we compete with any policy whose value function can be realized by $\mathcal{Q}$.*

The proof is similar to that of Proposition 10 and hence omitted.

Figure 7: Expected return and IPM of the learned policies as a function of training iterations in the policy optimization experiments. Results in this figure are averaged over 5 random seeds. **Top row**: Behavior policy is $\tau = 0.1$; **Bottom row**: Behavior policy is $\tau = 1.0$.

## J.4 Connection between MUB-PO and OLIVE [30]

A complete algorithm for exploration usually involves multiple iterations data collection and policy re-computation, and we only show that MUB-PO performs one such iteration effectively. There are further design choices needed to complete MUB-PO into a full algorithm. For example, one may repeatedly collect new data using the policy computed by MUB-PO and merge it with the data from previous rounds. However, it is difficult to analyze the algorithm theoretically: the realizability of $\mathcal{C}(\mathcal{W})$ depends on $\mu$, but $\mu$ itself dynamically changes over the execution of the algorithm due to data pooling, and any realizability-type assumptions such as $w_{\pi/\mu} \in \mathcal{W}$ are no longer static and cannot appear in an *a priori* guarantee.

One interesting way to avoid this difficulty is to use a special class of importance weights $\mathcal{W}$: instead of $w$ that depends on $(s, a)$, consider $w$ that is $(s, a)$-independent and is indicator function of the *identity of the dataset*. That is, the $\sup_{w \in \mathcal{W}}$ in $\mathrm{UB}_w^\pi$ chooses among the datasets collected in different rounds, and takes expectation w.r.t. only one of them (without reweighting within the dataset), essentially avoiding data pooling. The resulting algorithm is very similar to a parameter-free variant[16] of the OLIVER algorithm by Jiang et al. [30, Algorithm 3], which has been shown to enjoy low-sample complexities in a wide range of low-rank environments.

## K  Policy Optimization Experiments

In this section, we report the policy optimization results in CartPole environment. Due to the difficulty of optimization, we follow Nachum et al. [20] and consider the following simplified heuristic version

of MUB-PO and MLB-PO, which mostly applies in near-deterministic environments:

$$\arg\max_\pi \max_q q(s_0, \pi) - \beta\mathbb{E}_\mu[|r + \gamma q(s', \pi) - q(s, a)|] \qquad \text{(Simplified MUB-PO)}$$

$$\arg\max_\pi \min_q q(s_0, \pi) + \beta\mathbb{E}_\mu[|r + \gamma q(s', \pi) - q(s, a)|] \qquad \text{(Simplified MLB-PO)}$$

Here $\beta$ is a hyperparameter (similar to the role of $\alpha$ in the OPE experiments) that controls the level of pessimism in MLB-PO and optimism in MUB-PO. We parameterize the policy $\pi(\cdot|s)$ using a $32\times32$ MLP with a softmax layer.

For MUB-PO, we update $q$ and $\pi$ iteratively every 500 iterations. As for the lower bound, in order to avoid the policy becoming greedy so quickly, we only update $\pi$ once before alternating to $Q$. Besides, we also report the results of DQN [36] for comparison. Among all the experiments, we fix the learning rate as 5e−3 and the batch size as 500. All the Q-functions are parameterized by a $32\times32$ MLP.

Moreover, we also record the IPM of each algorithms during training, defined as:

$$\text{IPM} = \max_{f\in\mathcal{F}} \left| \mathbb{E}_{(s,a)\sim d^{\pi_b}}[f(s, a)] - \mathbb{E}_{(s,a)\sim d^\pi}[f(s, a)] \right|.$$

The IPM score measures the difference of the state-action visitation between the behavior policy and the policy we are optimizing, which is inspired by Proposition 14 and reflects whether $\pi$ explores new states and actions outside the coverage of $d^{\pi_b}$. In practice, $\mathcal{F}$ is constructed by 50 randomly initialized $32\times32$ MLPs.

In Fig. 7, we report the results with different behavior policies: $\tau = 0.1$ (behavior policy is nearly deterministic) and $\tau = 1.0$ (behavior policy is more exploratory). For MLB-PO and MUB-PO, we use $\beta = 1/(1 - \gamma)$ and $\beta = 3/(1 - \gamma)$ in $\tau = 0.1$ and $\tau = 1.0$, respectively. In both cases, MLB-PO delivers good performance as expected, and when data is non-exploratory ($\tau = 0.1$) it outperforms DQN. Notably, MLB-PO does not use explicit regularization towards behavior policy as many popular algorithms do. MUB-PO, on the other hand, fails to obtain a policy of high return, but the IPM scores show that it can induce an occupancy significantly different from $d_{\pi_b}$, which is what our theory anticipated (Proposition 14).

## L   Handling Statistical Errors based on Generalization Error Bounds

In this section we briefly explain how to construct a confidence bound that is valid with high probability in the presence of sampling errors due to only having access to a finite dataset. We emphasize that the construction based on generalization error bounds is typically loose for practical purposes and this section only serves as a conceptual illustration. We will use $\text{UB}_\text{w} := \inf_{w\in\mathcal{W}} \sup_{q\in\mathcal{Q}} L(w, q)$ as an example and the other upper/lower bounds can treated similarly. Let $\widehat{\text{UB}_\text{w}}$ denote the sample-based approximation of $\text{UB}_\text{w}$:

$$\widehat{\text{UB}_\text{w}} = \inf_{w\in\mathcal{W}} \sup_{q\in\mathcal{Q}} \hat{L}(w, q) := q(s_0, \pi) + \mathbb{E}_D[w(s, a)(r + \gamma q(s', \pi) - q(s, a))].$$

where $D$ is a dataset with $n$ i.i.d. data points $(s, a, r, s')$ sampled as $(s, a) \sim \mu, r \sim R(s, a), s' \sim P(s, a)$. We assume $\mathcal{W}$ and $\mathcal{Q}$ have bounded ranges of output values, that is, there exists two constants $C_\mathcal{W}, C_\mathcal{Q} > 0$, such that, $w \in \mathcal{W}, q \in \mathcal{Q}, \|w\|_\infty \leq C_\mathcal{W}, \|q\|_\infty \leq C_\mathcal{Q}$. As a result,

$$|L(w, q)| \leq |q(s_0, \pi)| + \mathbb{E}_\mu[|w(s, a)|(r + \gamma|q(s', \pi)| + |q(s, a)|)]$$

$$\leq C_\mathcal{Q} + \frac{C_\mathcal{W}}{1 - \gamma}(1 + (1 + \gamma)C_\mathcal{Q}) := L_{\max}.$$

In the following, we will use $L_{\max}$ as a shorthand for the upper bound of $|L(w, q)|$.

According to standard results for Rademacher complexity, we have with probability $1 - \delta$, for any $w \in \mathcal{W}$ and $q \in \mathcal{Q}$,

$$L(w, q) \leq \hat{L}(w, q) + 2\hat{\mathcal{R}}_D(\mathcal{W}, \mathcal{Q}) + 6L_{\max}\sqrt{\frac{\log\frac{2}{\delta}}{2n}}. \qquad (22)$$

where $\hat{\mathcal{R}}_D(\mathcal{W}, \mathcal{Q}) := \mathbb{E}_\sigma[\sup_{(w,q) \in \mathcal{W} \times \mathcal{Q}} \frac{1}{n} \sum_{(s,a,r,s') \in D} \sigma_i L^{(s,a,r,s')}(w,q)]$ is the empirical Rademacher complexity of $\mathcal{W} \times \mathcal{Q}$ w.r.t. sample $D$, which can be computed from data.

We define $(w^*, q^*) = \arg\min_{w \in \mathcal{W}} \max_{q \in \mathcal{Q}} L(w,q)$ and $(\hat{w}^*, \hat{q}^*) = \arg\min_{w \in \mathcal{W}} \max_{q \in \mathcal{Q}} \hat{L}(w,q)$ and use $q_{\hat{w}^*}$ to denote $\arg\max_{q \in \mathcal{Q}} L(w,q)(\hat{w}^*, q)$. Then, we have:

$$\text{UB}_\text{w} = L(w^*, q^*) \leq L(\hat{w}^*, q_{\hat{w}^*}) \tag{23}$$

$$\leq \hat{L}(\hat{w}^*, q_{\hat{w}^*}) + 2\hat{\mathcal{R}}_D(\mathcal{W}, \mathcal{Q}) + 6L_{\max}\sqrt{\frac{\log \frac{2}{\delta}}{2n}}$$

$$\leq \hat{L}(\hat{w}^*, \hat{q}^*) + 2\hat{\mathcal{R}}_D(\mathcal{W}, \mathcal{Q}) + 6L_{\max}\sqrt{\frac{\log \frac{2}{\delta}}{2n}}$$

$$= \widehat{\text{UB}}_\text{w} + 2\hat{\mathcal{R}}_D(\mathcal{W}, \mathcal{Q}) + 6L_{\max}\sqrt{\frac{\log \frac{2}{\delta}}{2n}}. \tag{24}$$

Eq.(24) implies that, for a fixed policy $\pi$, with probability $1-\delta$, when $\text{UB}_\text{w}$ is a valid upper bound (for example, when $Q^\pi$ is realizable), then we can guarantee that $\widehat{\text{UB}}_\text{w} + 2\hat{\mathcal{R}}_D(\mathcal{W}, \mathcal{Q}) + 6L_{\max}\sqrt{\frac{\log \frac{2}{\delta}}{2n}}$ is also a valid upper bound.

## M   Variants of Minimax Value Interval: Using Knowledge of Behavior Policy and Finite-horizon / Average-reward Case

While we derive the intervals in infinite-horizon discounted MDPs under the behavior-agnostic setting in the main paper, our derivations can be easily extended to other settings by replacing Lemmas 1 and 4 with their counterparts and following the same recipe. Below we briefly introduce these extensions.

### M.1   Using Knowledge of the Behavior Policy

When the behavior policy that produces the data is known (e.g., $\pi_b$ such that $\mu(s,a) = \mu(s)\pi_b(a|s)$), one can derive similar intervals that use importance weighting over actions $\rho := \frac{\pi(a|s)}{\pi_b(a|s)}$ with the following lemmas in place of Lemmas 1 and 4:

**Lemma 16.** *For any $w : \mathcal{S} \to \mathbb{R}$,*

$$J(\pi) - \mathbb{E}_w[\rho r] = V^\pi(s_0) + \mathbb{E}_w[\gamma \rho V^\pi(s') - V^\pi(s)]. \tag{25}$$

**Lemma 17.** *For any $v : \mathcal{S} \to \mathbb{R}$,*

$$J(\pi) - v(s_0) = \mathbb{E}_{d^\pi}[r + \gamma v(s') - v(s)]. \tag{26}$$

Following the recipe in Sec. 4 and 5, one can prove the properties of the corresponding intervals that are analogues to the results in this paper. Note that we can also use these lemmas to derive the counterparts of MWL and MQL for state-functions, the former of which is a variant of Liu et al. [5]'s method; see Uehara et al. [1, Appendix A.5 and Footnote 15].

### M.2   Finite-horizon Case

Finite-horizon MDPs can be modeled as infinite-horizon discounted MDPs with initial state distributions (which is our setting) by including time-step as a part of the state representation, modeling termination as absorbing states, and setting $\gamma = 1$.

### M.3   Average-reward Case

The average-reward case is somewhat more different from the discounted or the finite-horizon setting, so we go over it in more details. Suppose the Markov chain induced by $\pi$ is ergodic and let $d^\pi$ be its stationary state-action distribution (note that $d^\pi$ here is normalized, unlike the discounted occupancy

in the discounted setting). Similarly we define $w_{\pi/\mu}(s, a) := \frac{d^\pi(s,a)}{\mu(s,a)}$, and the quantity of interest is

$$J(\pi) = \lim_{T \to \infty} \frac{1}{T} \sum_{t=0}^{T-1} r_t = \mathbb{E}_{d^\pi}[r] = \mathbb{E}_w[r]. \tag{27}$$

As before, we need two Bellman equations, one for the distribution and the other one for the value function. The one for distribution is simple: for any $q : \mathcal{S} \times \mathcal{A} \to \mathbb{R}$,

$$\mathbb{E}_{d^\pi}[q(s, a) - q(s', \pi)] = 0. \tag{28}$$

Recall that $\mathbb{E}_{d^\pi}[\cdot]$ means taking expectation over $(s, a) \sim d^\pi, r \sim R(s, a), s' \sim P(s, a)$. This equation is just a re-expression of the fact that $d^\pi$ is the stationary distribution induced by $\pi$.

The Bellman equation for the value function is: for any $w : \mathcal{S} \times \mathcal{A} \to \mathbb{R}$,

$$\mathbb{E}_w[Q^\pi(s, a) + J(\pi) - r - Q^\pi(s', \pi)] = 0. \tag{29}$$

Since there is no initial state distribution in the average-reward case, even if $Q^\pi$ is learned, one cannot plug-in initial state distribution to obtain the estimation of $J(\pi)$. Instead, $J(\pi)$ itself directly appears as a separate quantity in the Bellman equation. In fact, value functions in the average-reward case are actually "difference of values", which are analogues to the advantage function in the discounted case and can be informally defined as

$$Q^\pi(s, a) = \mathbb{E}\left[\sum_{t=0}^{\infty}(r_t - J(\pi)) \,\middle|\, s_0 = s, a_0 = a, a_{1:\infty} \sim \pi\right].$$

According to this definition, the "temporal difference" $Q^\pi(s, a) - r - Q^\pi(s', \pi)$ is not zero in expectation, and a $J(\pi)$ term remains, allowing $J(\pi)$ to be directly manifested in the Bellman equations.

With the above background, we are ready to state the counterparts of Lemmas 1 and 4 for the average-reward case:

**Lemma 18.** *For any $w : \mathcal{S} \times \mathcal{A} \to \mathbb{R}$,*

$$J(\pi) = \mathbb{E}_w[r + Q^\pi(s', \pi) - Q^\pi(s, a)]. \tag{30}$$

**Lemma 19.** *For any $q : \mathcal{S} \times \mathcal{A} \to \mathbb{R}$,*

$$J(\pi) = \mathbb{E}_{w_{\pi/\mu}}[r + q(s', \pi) - q(s, a)]. \tag{31}$$

The resulting intervals are $\{\inf_w \sup_q, \sup_w \inf_q, \inf_q \sup_w, \sup_q \inf_w\} L(w, q)$, where $L(w, q) := \mathbb{E}_w[r + q(s', \pi) - q(s, a)]$. We note that the loss $L(w, q)$ is highly similar to the primal-dual form of LP for MDPs [37].