[Reviews · NeurIPS 2020]

Review 1

Summary and Contributions: #### Post author feedback #### The authors address my concerns in their rebuttal. Therefore, I increase my score to 7. ########################## The paper studies off-policy policy evaluation in reinforcement learning with function approximation. It unifies two styles of learning “value learning and weight learning” via the introduced minimax confidence interval (CI) method and it characterizes the bias due to the approximation errors. Finally, the paper provides experimental results to support its theoretical findings in both policy evaluation and policy optimization setting.

Strengths: The paper devises by simple derivations the minimax confidence interval approaches. It is closely connected to previous methods introduced in prior works such as MQL, MWL and AlgaeDICE but the paper does a good job to unify them. The author offer also insight on the behavior of CI in the misspecification setting (for both the marginalized importance weights and the Q-value) by characterizing the validity and the tightness of the CI. The paper also conducted a set of experiments to illustrate some theoretical predictions.

Weaknesses: The empirical part of the paper is limited. I can understand that maybe the focus of the papier is rather theoretical but it would be nice to improve the empirical part. In particular, in section 4.4 it would be nice to see the effect of the expressive power of W on the the validity and the tightness of CI. In Figure 1, could you also explain why we tend to have larger CI with large Q network and why the CI tend to exclude slightly the ground truth with moderately large Q class ? In figure 3, the validity ratio is not defined before, how is it computed ? For the policy optimization, in figure 2. MUB-PO fails totally. Authors explains this by the fact that MUB-PO may induce state distribution that is different from the training distribution. They also relate to MUB-PO to R_max, a classical algorithm designed for exploration. It would be very great to run MUB-PO is online setting to assess its exploration capability as claimed by the paper.

Correctness: I went through derivation and proofs but not in details. it seems correct to me

Clarity: the paper is overall well-written. The conclusion or discussion about future work is missing. it is bit direct to finish the paper by an experimental sub-section. Minor: - in 175 UB_w and LB_w should be UB_q and LB_q ?

Relation to Prior Work: The paper is closely related to MQL and MWL in Uhera et al 2019 and Algaedice et al in Nachum et al 2019. I understand that there is page limit but I think the paper needs a background section to describe MQL and MWL and Algaedice algorithms. Especially, the paper uses different notations than Algaedice's paper, it is hard to parse the two papers to find out the connections between each methods. In addition, I suspected that MUB-PO is the same as Frenchel Algaedice when the regularization coefficient \alpha in Algaedice is negative (maybe with \alpha=-1). Could the authors comment on this please ?

Reproducibility: Yes

Additional Feedback: I am willing to increase my score if the authors address some of my comments.


Review 2

Summary and Contributions: #### Post author feedback #### My main concern regarding the paper was not treating the sampling error. The authors convinced me in their response that it is a difficult problem and I don't think not having solved that should prevent acceptance of this paper. I have therefore increased my score to 7. ########################## The paper proposes a way to bound the off-policy evaluation errors when using one of the recently popular family of methods which estimate the stationary state distribution (or its importance sampling weights with respect to a dataset). The authors do this by deriving lower an upper bounds for estimation with respect to optimization of a function class approximating either weights or value function. The authors unify the two approximation methods by showing their equivalence as a min-max optimization problem.

Strengths: The analysis method is fairly straightforward, but manages to give a strong intuition for understanding the uncertainty of the OPE estimators considered which stems from approximation difficulties and model misspecification. The unification of optimization in the space of weights/q-function is interesting, and so overall the paper makes a nice contribution to understanding the uncertainties of this class of OPE methods.

Weaknesses: While the paper in its present form is already (marginally) good enough for publication, it still feels a bit preliminary and I would have preferred to see it published in a more mature version. I think the biggest contribution would come from considering the uncertainty introduced by sampling noise. The authors mention that they leave that for future work, but the paper would feel much more complete if this source of uncertainty would be considered, as it can be significant in practice. The empirical results could be strengthened with more domains (and more complicated) rather than just cart-pole. While I think this would improve the paper, this is not a major weakness as there are still enough results to demonstrate the ideas presented in the paper.

Correctness: The paper is correct to the extent I checked.

Clarity: The paper is written clearly with minor typos which should be polished.

Relation to Prior Work: The paper properly places itself in the context of prior work.

Reproducibility: Yes

Additional Feedback: This maybe an issue of taste, but I really don't like the abuse of the term "CI" for the bounds the authors are computing. I think it's misleading both in the sense that it does not consider the sampling noise, and doesn't have any statistical interpretation, and both of these elements are very characteristic of what I think most people usually think about when talking about CIs. i would have felt much more comfortable with a term like "approximation error bound" or something similar.


Review 3

Summary and Contributions: #### Post author feedback #### The authors partially address my concerns in their rebuttal. I still think the experimental should be strengthed largely. Therefore, I keep my score to 6. ########################## This paper mainly focuses on the off-policy evaluation for minimax methods that use value function and importance sampling. The authors develop the interval confidence interval under several circumstances. The interval confidence also connect two types of methods, weight learning and value learning. In the end, empirical results are used to show the validation of the proposed interval.

Strengths: The paper investigates an important problem (off policy evalution) from the perspective of theortical confidence interval. As we see great success by using RL in games and real applications, theortical analysis of the performance and evalution is needed. Therefore, I personaly think this paper is a good try.

Weaknesses: The emperiment scenerios are not rich to show the implication of the proposed confidence interval. More experiments can be added.

Correctness: The logic is sound and supported by experiments.

Clarity: The paper is easy to follow and clearly written.

Relation to Prior Work: Related works are well addressed.

Reproducibility: Yes

Additional Feedback:


Review 4

Summary and Contributions: This paper studies confidence interval (no uncertainty quantification but quantifies the bias due to the use of function approximations) for off-policy evaluation based on marginalized importance sampling. Due to the recent success of marginalized importance sampling for OPE problem, it is worth to understand deeper from a theoretical perspective.

Strengths: I feel this paper provides us deeper understanding on function approximation for Q-function and marginalized importance weights. When either Q-function and marginalized importance weights are realizable in two function class, this paper provides a quantification on how the bias affects the estimation.

Weaknesses: The study of bias issue is important, but I am not fully convinced the motivation of this so-called "confidence interval". Normally the confidence interval is designed for uncertain quantification and thus of great practical interest. However, although the authors explicitly point out they do not consider uncertainties, this will rule out all the important applications that typical CI could do (safe RL or else) (this CI will not be valid in practice due to estimation error). Thus, I can only view the contribution in this paper as sort of additional guarantee for the algorithm proposed in "Minimax Weight and Q-Function Learning for Off-Policy Evaluation" since the algorithms are the same. Solely quantifying a bias of an existing estimator may not be viewed as sufficiently significant. It is better to discuss after we identify those bias, can I reduce the bias in some way and improve existing algorithms? The use of the derived CI for policy optimization is not very clear. Without considering the uncentaities from the sample, the CI is not vaild for practical use. Since this is batch setting, where does exploration/exploitation come from? This needs to be clarified since commonly exploration/exploitation refers to online setting. The authors argue that the contributions over Liu et al. [5], Uehara et al. [1] are to make effective use of all components of data. This argument is not strongly supported in the following context (in theory or experiments). Overall, there are many pieces of merits in this paper but may lack a strong point to support the contribution over Uehara et al. [1].

Correctness: I think so.

Clarity: Generally good. Section 5 may need some improvements.

Relation to Prior Work: Good enough and clear.

Reproducibility: No

Additional Feedback:

[Author Response · NeurIPS 2020]

We thank the reviewers for their comments, and are very glad to see that all reviewers appreciate the novel and interesting
insights that are clearly conveyed by our presentation and simplified derivations (which we believe to carry pedagogical
values). They also raise some great points and criticisms, which we respond to below.

**[All] More Experiments**  All reviewers want more experiments, though R1 "understands that the focus of the paper
is rather theoretical" and R2 thinks "this is not a major weakness as there are still enough results to demonstrate the
ideas presented in the paper". While we have many interesting theoretical results (some of which were deferred to
the appendix due to page limit, e.g., App. B, E, the nontrivial proof of Prop.8, etc.) and accompanied them with
proof-of-concept experiments, it is hard to deny that more experiments would always help! Here are what we can do:
• We can show how CI changes with $\mathcal{W}$, as suggested by R1.
• We can run the experiments on more domains (probably in the appendix, as we want the main text to focus on theory).
• The minimax objective is difficult to optimize in practice, which is a common and somewhat unsolved issue in this line
of work. We have tried many methods to stablize training and documented the working tricks in F.2. In fact, we have
also conducted additional experiments on a tabular environment to understand how the behavior of our optimization
procedure deviates from an "ideal one" (which we can calculate only in simple environments). We will add these results
to the appendix. We find them more insightful than simply running the same experiments on more domains, especially
given that the community's understanding in the optimization aspect of these algorithms is still rather immature.

**[R2] Abuse of "CI"**  Agreed. We are considering changing "confidence interval" to "value interval" to avoid this issue.

**[R2] Sampling Errors**  We fully agree that addressing this issue is of high practical significance. In fact, we **do** handle
it in the experiments (see below). While we can add related discussions to make the paper conceptually more complete,
we also genuinely think that handling sampling errors in a way that is ***simultaneously*** theoretically sound and practically
useful is highly nontrivial (we have thought about it for quite a while) and may require an entire follow-up paper (or
more) and possibly some breakthroughs in theory.
• [Theory] We can handle sampling errors by adding e.g., Rademacher complexity-based generalization error bounds
(similar to Thm 9 of Uehara et al) to the interval, which is straightforward from textbook. It makes the paper's theory
more complete, but as we all know, even the tightest generalization bounds are often too loose to be practically useful.
• [Experiments] **We do handle sampling errors in the experiments via bootstrapping** (Fig 3), which is often the
most practical option when strict CIs tend to be loose. Even so, there are still many open questions: for example,
the difficulty of optimization—which is serious in this setting—could affect the CI's validity. In light of this, maybe
we should keep the outer player fixed (e.g., $w$ in $\inf_w \sup_q$), and only bootstrap the inner optimization for a more
conservative interval? How does this affect the theoretical properties of bootstrapping? After all, confidence intervals
for such minimax statistics (over neural nets!) are not something that classical statistics literature usually handle, which
means a lot need to be done in theory. Of course, maybe this is just our ignorance and pointers are always welcomed!
• While we leave sampling error to future work, we expanded the paper's scope by discussing policy optimization under
insufficient data (Section 5). This is a highly important yet understudied problem (L228), and the results we derived for
OPE were nicely applied to this problem, producing novel insights that are useful for future research and discussions.

**[R1] Describe MWL/MQL**  We have 2 "nutshell" paragraphs in L114 and 166, which compactly summarizes the core
ideas of MWL/MQL and foreshadows our derivations. If you think there are other aspects of MWL/MQL that the
readers need to know to understand our work, please let us know and we are happy to integrate them into the paper.

**[R1] AlgaeDICE**  The policy evaluation component of Fenchel AlgaeDICE (their Eq.16, with $\alpha = 0$ and without
"$\max_\pi$") is precisely our $\mathrm{UB}_q$ ($= \mathrm{LB}_w$ with convex classes): their $\nu$ is our $q$, their $\zeta$ is our $w$, and there are some
superficial differences due to e.g., normalization conventions; we will explain in detail in the revision.

We don't think MUB-PO can be recovered by negative $\alpha$. Your guess is probably based on the first line of their Eq.15,
$\max_\pi \mathbb{E}_{d^\pi}[r] - \alpha D_f(d^\pi \| d^D)$, so negative $\alpha$ should encourage exploration. However, this expression is only equal to
the next line when *unrestricted* function spaces for $\zeta$ and $\nu$ are used (which they implicitly assumed throughout their
derivations), which does not hold in general. Moreover, the $\alpha$ term can be viewed as regularization (see our App.E),
but MUB-PO achieves exploration even *without* regularization. The key of MUB-PO is $\inf_w \sup_q$, which differs from
$\sup_w \inf_q$ in MLB-PO/AlgaeDICE in a fundamental way. In fact, MUB-PO enjoys an exploration guarantee (Prop.10),
but it's unclear if AlgaeDICE with negative $\alpha$ enjoys the same type of guarantees under similar assumptions.

**[R1] Validity Ratio**  Our bad. It's the relative frequency that the groundtruth is contained in the predicted interval.

**[R1] Fig 1**  First, inclusion of groundtruth is only guaranteed when one of the classes is realizable, data is unlimited,
and optimization is exact. Since all conditions only hold approximately in practice, slight exclusion is expected. Now
about larger CI with larger $\mathcal{Q}$: when $\mathcal{W}$ is fixed, increasing $\mathcal{Q}$ reduces $\inf_q \sup_w$ and increases $\sup_q \inf_w$, that is, the
red bar in Fig 1 should monotonically decrease, and the blue bar increase, which is roughly the case. Therefore, once
the CI reversed ($Q$-net size $\geq 10$), increasing $\mathcal{Q}$ would only make the interval larger. The interval would only collapse
to a point and stay at 0 length with realizable $\mathcal{W}$ among other idealized assumptions.

[Meta-Review · NeurIPS 2020]

The paper provides a very general minimax framework for quantifying the bias/approximation error in off-policy evaluation, and the results apply to a range of OPE methods. Reviewers generally agree that this is a good paper and there is contribution. One potentially improvable direction would be to quantify the statistical noise in off-policy evaluation, which is nontrivial but extremely important. Reviewers, AC and SAC also agree that such analysis could be left for future work. We would also like to strongly suggest that the authors consider rephrase/explain the wording "confidence interval". In statistics, CI is mainly used to quantifying statistical error rather than approximation error. The current paper uses the word "CI" as as form of approximation error, but does not given statistical error analysis. Such use of "CI" could lead to potential misunderstanding, and should be clarified in the abstract and intro.